# LiveOIBench: Can Large Language Models Outperform Human Contestants in Informatics Olympiads?

**Kaijian Zou** [1]   **Aaron Xiong** [1]   **Yunxiang Zhang** [1]   **Xinliang Frederick Zhang** [1]   **Yueqi Ren** [1]   **Jirong Yang** [1]
**Ayoung Lee** [1]   **Shitanshu Bhushan** [1]   **Lu Wang** [1]

## Abstract

Competitive programming problems are increasingly used to evaluate the coding capabilities of large language models (LLMs) due to their complexity and ease of verification. Yet, current coding benchmarks face limitations such as a lack of exceptionally challenging problems, insufficient test case coverage, and reliance on online platform APIs that limit accessibility. To address these issues, we introduce LiveOIBench, a large-scale competitive programming benchmark featuring 403 expert-curated problems, averaging 60 official test cases each, drawn from 72 contests across 14 Informatics Olympiads held between 2023 and 2025. LiveOIBench has four key features: (1) expert-designed tasks with detailed subtask rubrics and extensive test cases; (2) direct comparison to elite human contestants; (3) continuous updates to reduce contamination risk; and (4) a fully offline, reproducible evaluation system. Benchmarking 34 popular general-purpose and reasoning LLMs, we find that GPT-5 achieves an 81.76th percentile, still falling short of top human contestants, while among the open-weight models, GPT-OSS-120B reaches only the 60th percentile. Reasoning-trace analyses indicate that robust reasoning models prioritize precise problem analysis over excessive exploration. Finally, analyses across release dates, task familiarity, and code similarity find minimal evidence of data contamination in our benchmark. Our leaderboard, code, and data are available at: https://liveoibench.github.io/

[1]University of Michigan, Ann Arbor. Correspondence to: Kaijian Zou <zkjzou@umich.edu>.

*Proceedings of the 43rd International Conference on Machine Learning*, Seoul, South Korea. PMLR 306, 2026. Copyright 2026 by the author(s).

## 1. Introduction

Coding has emerged as a critical domain for LLMs (Lai et al., 2023; Zhuo et al., 2025; Liu et al., 2024; Jimenez et al., 2024; Chan et al., 2025), with coding benchmarks serving as essential tools to evaluate LLMs' algorithmic reasoning capabilities as these models continue advancing through inference-time scaling techniques (Li et al., 2022a; Kojima et al., 2023; OpenAI et al., 2024; DeepSeek-AI et al., 2025; Li et al., 2025b). However, rapid improvements in model capabilities have led to saturation of traditional coding benchmarks such as HumanEval (Chen et al., 2021) and MBPP (Austin et al., 2021), prompting the adoption of competitive coding benchmarks (Hendrycks et al., 2021; Li et al., 2022a; 2023; Shi et al., 2024) such as LiveCodeBench (Jain et al., 2025) and CodeELO (Quan et al., 2025), which leverage problems from platforms like Codeforces for their complexity and ease of verification. Despite their strengths, these benchmarks have notable weaknesses: (1) overestimation of LLMs' performance due to high false-positive rates using incomplete test suites (Li et al., 2022a; Liu et al., 2023; Jain et al., 2025), (2) insufficient difficulty granularity and a lack of exceptionally challenging questions (Jain et al., 2025; Quan et al., 2025), (3) usage of external APIs for evaluation, restricting reproducibility and accessibility (Jain et al., 2025; Quan et al., 2025; Zheng et al., 2025; Li et al., 2025c), (4) reliance on coarse pass rates as the sole evaluation metric, which overlooks insights into nuanced model capabilities (Li et al., 2022a; Jain et al., 2025; Shi et al., 2024; Wang et al., 2026), and (5) costly updates due to the extensive human annotations and computational resources required (Wang et al., 2026; Zhu et al., 2025).

To address these gaps, we introduce LiveOIBench, the first comprehensive Informatics Olympiad coding benchmark constructed directly from official contest sources, featuring expert-designed test cases and official contestant rankings, and it is publicly available to support reproducible evaluation along with fine-grained scoring rubrics. Compared to previous benchmarks (Hendrycks et al., 2021; Li et al., 2022a; Jain et al., 2025; Shi et al., 2024; Quan et al., 2025) and concurrent work (Li et al., 2025c; Zheng et al., 2025; Zhu et al., 2025; Wang et al., 2026) in Table 1, LiveOIBench

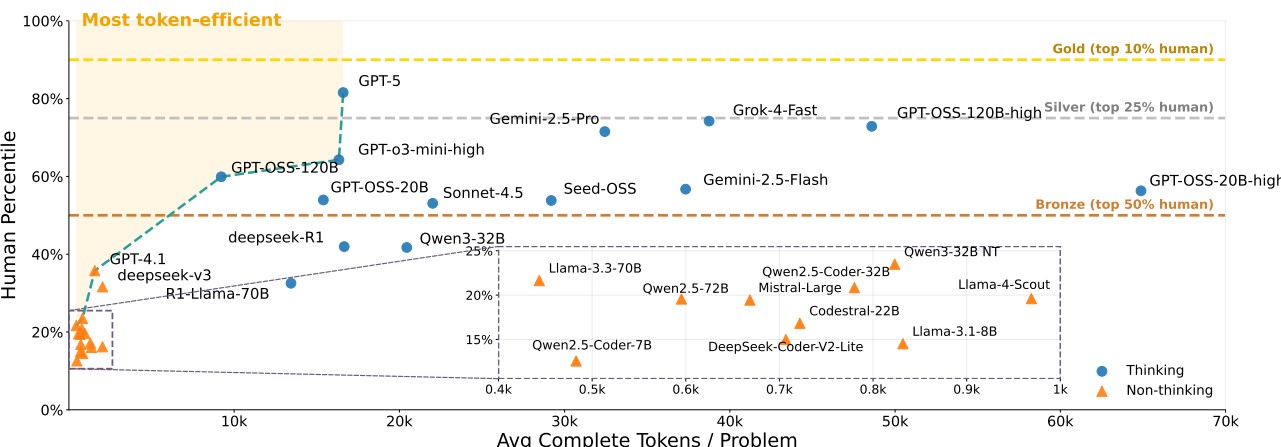

*Figure 1.* **LiveOIBench.** Average human percentile across all contests versus average completion tokens per problem. The dashed boxes highlight the lower performance range of non-thinking LLMs. OpenAI models lie on the token-efficiency frontier, achieving higher human percentile with fewer tokens. Despite improvements, all evaluated models remain below the Gold medal threshold (top 10% human performance), indicating substantial room for progress.

features the following key advancements:

1. **Expert-curated Tasks with Fine-grained Subtask Rubrics.** We source problems, test cases, and scoring rubrics from 14 Informatics Olympiads. This official suite eliminates high false-positive rates common in previous benchmarks and provides fine-grained subtask scoring for nuanced performance analysis.
2. **Direct Human Contestant Comparisons.** Official results from human competitors are collected, allowing direct benchmarking against human-level performance.
3. **Continuous, Contamination-free Updates.** Updates with newly released Olympiad tasks maintain benchmark freshness and minimize data contamination risks.
4. **Integrated Offline Evaluation System.** We develop a self-contained judge for fully offline, reproducible evaluations that removes reliance on external APIs and enhances research accessibility.

In total, LiveOIBench comprises 403 rigorously curated problems sourced from 72 contests across 14 Informatics Olympiads, each accompanied by an average of 60 expert-written test cases. Using LiveOIBench, we evaluate 34 leading models, revealing that proprietary models maintain a substantial performance advantage. In particular, GPT-5 (OpenAI, 2025b) achieves an average human percentile of 81.76, while also exhibiting remarkable token efficiency by reaching this performance with fewer than 20K reasoning tokens, positioning it on the efficiency frontier (Figure 1). Among open-weight alternatives, Seed-OSS (ByteDance Seed Team, 2025) achieves the 54th percentile and Qwen3-32B (Yang et al., 2025a) reaches the 42nd percentile, both demonstrating significant performance gains from additional reasoning tokens. Additionally, GPT-OSS-120B (OpenAI et al., 2025) attains the 60th percentile, effectively narrowing the performance gap with GPT-5 and highlighting significant progress in open-weight model capabilities. Moreover,

examining performance across different algorithms reveals current models' weaknesses in algorithms like dynamic programming, which demand creative pattern recognition and recursive decomposition. Detailed reasoning trace analyses further reveal that high-performing models strategically allocate more tokens to focused analysis rather than excessive exploration, underscoring that carefully managed reasoning behaviors are crucial for robust performance on challenging tasks. Finally, we conduct an in-depth analysis of data contamination in our benchmark, finding minimal correlation between model performance and problem release dates, task familiarity, or solution familiarity.

In summary, we make the following key contributions:

- **Data**: Release a comprehensive, high-quality competitive coding benchmark with expert-crafted problems, full test suites, and integrated human contestant results.
- **Evaluation**: Provide a robust local evaluation framework with test cases and subtask rubrics, enabling accessible, fine-grained human-model comparisons.
- **Benchmarking Results**: Conduct extensive benchmarking and detailed performance analysis of 34 leading open-source and proprietary models.
- **Analyses**: Perform extensive analyses of model performance across diverse algorithms, reasoning trace, data-contamination effects, and solution submission outcomes.

## 2. Related Work

Early code generation benchmarks such as HumanEval (Chen et al., 2021) and MBPP (Austin et al., 2021) mainly focus on the basic Python programs and have long served as standard evaluations of LLM code generation capability. However, as the capability of LLMs evolves, simple benchmarks like HumanEval no longer satisfy the current

*Table 1.* Comparison with existing coding datasets. LiveOIBench consists of continuously updated competitive coding problems from recent Informatics Olympiads, spanning various difficulty levels. Unlike previous benchmarks that generated test cases using predefined rules or LLMs, LiveOIBench features expert-curated test cases sourced *directly* from official competition organizers. It also provides an accessible *offline* evaluation platform, detailed subtask rubrics for *fine-grained* assessment, and official human contestant rankings for precise *human-model* comparisons.

| Dataset | Difficulty | Updates | Expert Test Cases | Offline Eval | Subtasks | Human Percentile |
|---|---|---|---|---|---|---|
| HumanEval | ★ | ✗ | ✗ | ✓ | ✗ | ✗ |
| APPS | ★★ | ✗ | ✗ | ✓ | ✗ | ✗ |
| CodeContests | ★★★ | ✗ | ✗ | ✓ | ✗ | ✗ |
| TACO | ★★ | ✗ | ✗ | ✓ | ✗ | ✗ |
| LiveCodeBench | ★★ | ✓ | ✗ | ✓ | ✗ | ✗ |
| USACO | ★★★ | ✓ | ✓ | ✓ | ✗ | ✗ |
| CODEELO | ★★★ | ✓ | ✓(hidden) | ✗ | ✗ | ✓ |
| OI-Bench | ★★★★ | ✗ | ✓(unofficial) | ✗ | ✗ | ✓ |
| LiveCodeBench-Pro | ★★★★ | ✓ | ✓(hidden) | ✗ | ✗ | ✓ |
| HLCE | ★★★★ | ✗ | ✓(hidden) | ✗ | ✗ | ✓ |
| AetherCode | ★★★★ | ✓ | ✓ (unofficial) | ✗ | ✗ | ✗ |
| LiveOIBench (Ours) | ★★★★ | ✓ | ✓(official and public) | ✓ | ✓ | ✓ |

benchmarking needs. Researchers have therefore started developing more realistic and challenging benchmarks (Lai et al., 2023; Yin et al., 2023; Zhuo et al., 2025; Liu et al., 2024; Jimenez et al., 2024; Chan et al., 2025). Specifically, DS1000 (Lai et al., 2023) and ARCADE (Yin et al., 2023) consist of data science problems in Python. BigCodeBench (Zhuo et al., 2025) collects code generation tasks from Stack Overflow, which involves more complex instructions and diverse function calls. The SWE-Bench (Jimenez et al., 2024) takes one step further and tests models' ability to solve real-world GitHub issues. This line of work emphasizes evaluating LLMs' ability to effectively implement, debug, and reason through complex real-world coding tasks.

In addition to real-world application benchmarks, there is another line of work that focuses on **competitive programming benchmarks** (Hendrycks et al., 2021; Li et al., 2022a; Jain et al., 2025; Quan et al., 2025), which test models' reasoning ability on challenging coding tasks under specified time and memory constraints. Prior competitive programming benchmarks typically collect problems from online coding platforms such as Codeforces and AtCoder, which do not release private test cases. The lack of sufficient private test cases can cause many false-positive solutions (Li et al., 2022a; Liu et al., 2023). Li et al. (2022a) augment test cases by mutating existing test inputs. Liu et al. (2023) leverages both LLM-based and mutation-based strategies to augment test cases with predefined rules. Even with over 200 additional tests per problem, Li et al. (2022a) shows there is still a nearly 50% false-positive rate. Other work (Quan et al., 2025; Zheng et al., 2025; Li et al., 2025c) tries to solve this problem by creating an evaluation system to submit LLM-generated solutions directly to the Codeforces platform. Although this approach ensures that solutions are tested on the whole test set, its dependency on the online platform limits its accessibility to the research community, as large-scale evaluations involving thousands of submis-

sions can overload platform servers.

To address these limitations, we collect problems from the official websites of many informatics Olympiads around the world. Most informatics Olympiads release complete test sets that are curated carefully by the organizing committees. We are among the first works to leverage problems from different informatics Olympiads and evaluate models' performance against human contestants. Prior research by Shi et al. (2024) exclusively used USACO problems with pass rate as the sole evaluation metric. Concurrent benchmarks, such as LiveCodeBench Pro (Zheng et al., 2025), HLCE (Li et al., 2025c), OI-Bench (Zhu et al., 2025), and AetherCode (Wang et al., 2026), also incorporate competitive programming tasks from sources like ICPC and IOI. However, LiveCodeBench Pro and HLCE primarily evaluate using Codeforces, limiting their accessibility. OI-Bench relies mostly on private, non-English school contests without continuous updates, while AetherCode relies on LLM-generated test cases and extensive human annotation, using pass rate as its sole evaluation metric. In contrast, our benchmark provides comprehensive coverage across diverse Olympiads, allows easy updates by directly collecting official test cases, and employs detailed evaluation metrics including subtask rubrics and human percentile comparisons.

## 3. LiveOIBench Construction

To construct LiveOIBench, we follow a clearly defined, step-by-step process combining automated data collection methods with manual verification to ensure dataset quality. In this section, we summarize the major steps in constructing our benchmark. A more detailed description of our construction process is provided in Appendix A.

**Competition Selection and Task Collection:** We first curate a comprehensive list of globally recognized interna-

tional Informatics Olympiads and select national contests from top-performing IOI countries where English task statements are available (see Table 8). For each selected contest, we develop a custom crawler that systematically extracts English task statements (see Appendix A.6) directly from official competition websites, capturing details such as time and memory constraints, subtask specifications, test cases, official solutions, and contestant rankings. When official sites lack complete or up-to-date information, we supplement the data by retrieving missing details from established online platforms such as CSES[1] and LibreOJ[2]. To maintain temporal relevance of our benchmark, we strictly limit our dataset to contests held in 2023 or later. Additionally, we provide full descriptions of each competition along with official websites in Appendix A.7, ensuring selected contests have extensive historical data, consistent participant numbers, and regularly hosted events. Using our crawlers, we will streamline our collection process for future contests and continuously update our benchmark by crawling monthly or annual problem releases from 14 actively maintained competition websites.

**Quality Assurance and Markdown Conversion:** After running the crawlers, we verify that all the data are downloaded properly and execute the official solutions from contest organizers to confirm the correctness of crawled test cases and the robustness of our evaluation judge. Given that many contests provide task statements exclusively as PDF documents, we employ Marker[3] to automatically convert these PDFs into markdown format. We further utilize Gemini-2.0-Flash to automatically verify and correct these markdown texts. To ensure conversion accuracy, we manually inspect a sample of 40 tasks before batch processing.

**Metadata Enrichment:** We enhance the dataset with supplementary metadata, including difficulty and algorithm tags such as dynamic programming and greedy, crawled from solved.ac[4] and Luogu[5]. Tasks and metadata are matched using competition dates, task titles, and problem identifiers.

**Contestant Matching and Codeforces Ratings:** Beyond raw human contestant results, contestants are automatically linked to their respective Codeforces profiles based on their names, user IDs, and countries, while contestants whose profiles cannot be confidently matched are skipped. Verified profiles are then queried via the Codeforces API to retrieve user ratings from 2022 to 2025.

Ultimately, LiveOIBench comprises **403 rigorously curated problems** from **72 contests** across **14 Informatics Olympiads**, conducted between 2023 and 2025. The bench-

mark statistics are detailed in Table 7 with competition information in Appendix A.7.

There are four characteristics that make our dataset challenging and unique compared to the existing coding datasets:

- **Challenging Problems with Subtasks.** Expert-curated problems contain subtasks with constraints, enabling precise evaluation through partial scoring.
- **Expert-Designed Test Cases.** Includes test cases curated carefully by the organizing committees rather than test cases generated by predefined rules or LLMs, ensuring evaluation free of false positives.
- **Direct Human Comparisons.** Benchmarks LLM performance against human contestants using percentile ranks, medals, and Codeforces Elo ratings.
- **Live Updates.** Continuously updated with recent contests to minimize data contamination. All 14 competitions in Appendix A.7 will be updated periodically.

## 4. Benchmarking Results

We evaluate a total of 34 LLMs. These models are categorized into three groups based on their accessibility and "thinking" capabilities: proprietary LLMs, open-weight thinking LLMs, and open-weight non-thinking LLMs. More details about models and prompts can be found in Appendix B and Appendix E.7. During inference, we sample 8 candidate solutions per model in C++ and select the solution with the highest score (Jain et al., 2025; Quan et al., 2025). We discuss a comparative analysis with Java and Python in Section 5.5, with detailed results in Appendix E.3. We adopt the following evaluation metrics: `pass rate` (Kulal et al., 2019; Chen et al., 2021), `relative score`, `human percentile`, `Olympiad medal system`, and `Codeforces Elo` (Quan et al., 2025; Zheng et al., 2025). With subtask rubrics and human contestant results, we can calculate each model's total points in a contest, allowing precise comparisons to human contestants via percentile rankings and medal awards. The description of each metric can be found in Table 2 or Appendix C. In Table 2, we present benchmarking results for selected models that rank top in their corresponding categories. Full results for all evaluated models are included in Table 12.

**Proprietary LLMs remain dominant, yet open-weight models are narrowing the performance gap.** Our findings indicate that proprietary LLMs continue to lead in competitive coding benchmarks. Specifically, GPT-5 achieves impressive results, securing gold medals in 50% of contests, winning medals of any type in 88.89% of contests, and outperforming an average of 81.76% of human contestants. Grok-4-Fast shows very competitive performance, with a human percentile of 74.23, ranking second-highest among all models. Claude-Sonnet-4.5 lags behind all other

---

[1]https://cses.fi

[2]https://loj.ac

[3]https://github.com/datalab-to/marker

[4]https://solved.ac

[5]https://www.luogu.com.cn

*Table 2.* Main results of best-performing models in each category evaluated on all 72 contests. Full results are presented in Table 12. **Gold** and **Medals**: % of contests in which a model achieved a gold medal or any medal, respectively. **Relative Score**: % of total contest points obtained by the model. **Human Percentile**: % of human contestants that a model surpasses. **Pass Rate**: % of tasks where a model successfully passes all test cases. **Elo**: the Codeforces Elo rating earned by a model based on performance relative to human contestants. All metrics are higher the better. Notably, the highest-performing GPT-5 achieves an impressive 81.76th percentile but still falls short of top human contestants, successfully solving only 63% of tasks in the benchmark.

| Model | 🥇 Gold (%) | 🏆 Medals (%) | 📋 Relative Score(%) | 👤 Human Percentile (%) | ✅ Pass Rate (%) | 📊 Elo |
|---|---|---|---|---|---|---|
| 🔒 Proprietary LLMs | | | | | | |
| GPT-5 | **50.00** | **88.89** | **67.21** | **81.76** | **63.03** | **2414** |
| Grok-4-Fast | 45.83 | 83.33 | 56.99 | 74.23 | 50.95 | 2221 |
| Gemini-2.5-Pro | 31.94 | 77.78 | 51.33 | 71.80 | 44.46 | 2192 |
| GPT-o3-Mini-High | 26.39 | 72.22 | 47.69 | 64.28 | 44.19 | 2088 |
| Gemini-2.5-Flash | 15.28 | 62.5 | 41.29 | 56.81 | 36.06 | 1945 |
| Claude-Sonnet-4.5 | 11.11 | 66.68 | 38.30 | 53.08 | 27.05 | 1848 |
| GPT-4.1 | 4.17 | 40.28 | 24.78 | 35.99 | 18.32 | 1482 |
| Open-weight Thinking LLMs | | | | | | |
| GPT-OSS-120B-High | **50.00** | 87.50 | 62.78 | 72.88 | 60.14 | 2205 |
| GPT-OSS-120B | 29.17 | 73.61 | 49.23 | 59.90 | 47.78 | 2032 |
| GPT-OSS-20B | 19.44 | 68.06 | 42.36 | 53.94 | 42.80 | 1901 |
| Seed-OSS | 15.28 | 68.06 | 42.58 | 53.81 | 40.09 | 1873 |
| Qwen3-32B | 9.72 | 54.17 | 32.86 | 42.00 | 27.70 | 1665 |
| DeepSeek-R1 | 6.94 | 52.78 | 33.43 | 42.29 | 28.87 | 1617 |
| Qwen3-14B | 5.56 | 45.83 | 27.24 | 34.59 | 22.73 | 1402 |
| DeepSeek-R1-Distill-Llama-70B | 1.39 | 33.33 | 20.50 | 32.30 | 16.88 | 1284 |
| Open-weight Non-Thinking LLMs | | | | | | |
| DeepSeek-V3 | 4.17 | 34.72 | 21.70 | 31.76 | 17.10 | 1283 |
| Qwen3-32B-Non-Thinking | 1.39 | 16.67 | 12.92 | 24.64 | 8.78 | 1040 |

thinking proprietary models, ranking only in the 53.08th percentile. Among open-source models tested, GPT-OSS-120B emerges as the strongest competitor. Under standard reasoning effort, GPT-OSS-120B achieves gold medals in 29.17% of contests and performs near the 60th percentile—approximately 21.86 percentile points below GPT-5. Notably, with high reasoning effort, GPT-OSS-120B surpasses Gemini-2.5-Pro and trails GPT-5 by merely 9 percentile points. Seed-OSS, the second-best open-source model, attains the 54th percentile, narrowly trailing Gemini-2.5-Flash by only 3 percentile points. However, other models exhibit substantial performance gaps, with Qwen3-32B and DeepSeek-R1 obtaining gold medals in only 10% and 7% of contests, respectively, and performing at roughly the 42nd percentile. Smaller and less powerful models, such as Qwen3-4B and DeepSeek-R1-Distill-Llama-8B, exhibit notably lower performance—Qwen3-4B secures gold medals in only 1.39% of contests, while DeepSeek-R1-Distill-Llama-8B achieves no gold medals and ranks only at the 11th percentile. These results clearly demonstrate that achieving meaningful performance on competitive programming tasks in LiveOIBench requires LLMs with substantial reasoning capabilities. **Even the leading GPT-5 model falls short of top-tier human contestants.** Achieving a gold medal in every contest requires consistently surpassing the 90th percentile. Although GPT-5 demonstrates remarkable capabilities with a near 82nd percentile and a rating

of 2414, its performance still lags behind elite human competitors. This highlights an ongoing challenge for LLMs in surpassing human expertise in competitive coding.

**Thinking models perform significantly better than non-thinking models.** Models lacking extended thinking capabilities perform notably worse in our benchmark. GPT-4.1, the highest-performing non-thinking model evaluated, achieves results comparable only to Qwen3-14B.

Apart from GPT-4.1 and DeepSeek-V3, all other non-thinking models fail to exceed a 10% pass rate, underscoring the critical importance of extended thinking in addressing complex competitive coding tasks. Extending this analysis, we investigate inference-time scaling techniques and find that both parallel (Chen et al., 2021; Jain et al., 2025) and sequential (Snell et al., 2025; DeepSeek-AI et al., 2025; Li et al., 2025a) scaling methods significantly enhance coding capabilities. In Figure 6a, parallel scaling identifies maximum coding capacity but shows diminishing returns beyond a few attempts. By increasing the reasoning budget, sequential scaling allows smaller models to approach larger-model performance in Figure 6b, reinforcing our earlier observation about the importance of extended thinking capabilities. For detailed analyses, see Appendix E.2.

**Comprehensive evaluation metrics provide deeper insights into model capabilities.** Relying solely on pass rate can obscure key aspects of model performance. For exam-

*Table 3.* Pass@8 of top-15 algorithm tags for selected models. Full results can be found in Table 13. Abbreviations: IM (implementation), MA (mathematics), AH (ad-hoc), PS (prefix sum), SO (sorting), GR (greedy), GTR (graph traversal), BS (binary search), NT (number theory), GT (graph theory), DS (data structures), CB (combinatorics), DP (dynamic programming), TR (tree), ST (segment tree). Darker color indicates the model performs better on this particular tag compared to other tags. Models generally perform better on algorithm tags that involve straightforward application of standard formulas or well-known patterns.

| Model | IM | MA | AH | PS | SO | GR | GTR | BS | NT | GT | DS | CB | DP | TR | ST |
|---|---|---|---|---|---|---|---|---|---|---|---|---|---|---|---|
| 🔒 **Proprietary LLMs** | | | | | | | | | | | | | | | |
| GPT-5 | 71.79 | 71.43 | 43.48 | 73.33 | 75.56 | 60.00 | 71.43 | 54.84 | 64.71 | 66.67 | 66.27 | 64.71 | 46.88 | 37.50 | 56.41 |
| Grok-4-Fast | 56.41 | 69.23 | 37.50 | 61.54 | 65.85 | 34.38 | 50.00 | 36.67 | 47.06 | 45.45 | 40.51 | 52.94 | 35.94 | 26.09 | 30.77 |
| Gemini-2.5-Pro | 66.67 | 71.43 | 30.43 | 53.33 | 57.78 | 37.14 | 42.86 | 38.71 | 35.29 | 44.44 | 38.55 | 58.82 | 23.44 | 20.83 | 30.77 |
| GPT-o3-Mini-High | 64.10 | 71.43 | 34.78 | 46.67 | 60.00 | 37.14 | 46.43 | 41.94 | 41.18 | 38.89 | 38.55 | 47.06 | 34.38 | 20.83 | 28.21 |
| Gemini-2.5-Flash | 64.10 | 71.43 | 30.43 | 46.67 | 48.89 | 28.57 | 25.00 | 32.26 | 29.41 | 29.63 | 30.12 | 47.06 | 20.31 | 12.50 | 15.38 |
| Claude-Sonnet-4.5 | 55.26 | 69.23 | 33.33 | 35.71 | 30.23 | 22.86 | 21.43 | 19.35 | 29.41 | 21.82 | 20.99 | 47.06 | 7.81 | 4.35 | 2.56 |
| GPT-4.1 | 53.85 | 50.00 | 26.09 | 40.00 | 13.33 | 14.29 | 7.14 | 12.90 | 17.65 | 12.96 | 12.05 | 29.41 | 6.25 | 4.17 | 5.13 |
| **Open-weight Thinking LLMs** | | | | | | | | | | | | | | | |
| GPT-OSS-120B | 64.10 | 64.29 | 34.78 | 53.33 | 60.00 | 40.00 | 53.57 | 38.71 | 41.18 | 44.44 | 44.58 | 58.82 | 35.94 | 25.00 | 35.90 |
| GPT-OSS-20B | 63.16 | 71.43 | 40.91 | 57.14 | 51.11 | 36.36 | 35.71 | 36.67 | 47.06 | 30.19 | 36.59 | 66.67 | 29.69 | 22.73 | 26.32 |
| Seed-OSS | 61.54 | 64.29 | 36.36 | 53.33 | 48.89 | 31.43 | 32.14 | 38.71 | 35.29 | 27.78 | 34.94 | 52.94 | 26.56 | 12.50 | 28.21 |
| Qwen3-32B | 58.97 | 61.54 | 30.43 | 35.71 | 28.89 | 21.88 | 21.43 | 16.67 | 29.41 | 22.64 | 22.22 | 29.41 | 14.29 | 4.35 | 8.11 |
| DeepSeek-R1 | 61.54 | 64.29 | 30.43 | 33.33 | 28.89 | 17.14 | 17.86 | 22.58 | 29.41 | 22.22 | 20.48 | 29.41 | 15.62 | 4.17 | 7.69 |
| Qwen3-14B | 51.28 | 64.29 | 25.00 | 42.86 | 25.00 | 17.14 | 14.29 | 12.90 | 29.41 | 18.18 | 19.51 | 35.29 | 12.50 | 4.17 | 5.13 |
| DeepSeek-R1-Distill-Llama-70B | 41.03 | 50.00 | 17.39 | 20.00 | 20.00 | 17.14 | 10.71 | 16.13 | 17.65 | 14.81 | 13.25 | 11.76 | 9.38 | 4.17 | 5.13 |
| **Open-weight Non-Thinking LLMs** | | | | | | | | | | | | | | | |
| DeepSeek-V3 | 51.28 | 46.15 | 21.74 | 28.57 | 20.00 | 12.50 | 14.29 | 13.33 | 17.65 | 15.09 | 14.81 | 11.76 | 7.94 | 8.70 | 8.11 |
| Qwen3-32B-Non-Thinking | 25.64 | 42.86 | 13.04 | 0.00 | 6.67 | 5.71 | 3.57 | 9.68 | 11.76 | 7.41 | 2.41 | 11.76 | 4.69 | 0.00 | 2.56 |

ple, GPT-OSS-120B achieves a higher pass rate (47.78%) compared to Gemini-2.5-Pro (44.46%); however, Gemini-2.5-Pro consistently surpasses GPT-OSS-120B in both human percentile ranking and Elo rating, indicating stronger overall competitive coding proficiency. We recommend that practitioners and researchers adopt a multifaceted evaluation approach: use Gold and Medals to gauge contest-level success, Human Percentile to contextualize model performance relative to humans, Elo to assess coding skill within the broader competitive coding community, and Pass Rate to evaluate core problem-solving capability. Utilizing these metrics collectively ensures a balanced and comprehensive understanding of model strengths and limitations.

**Later subtasks are more challenging.** We investigate how model performance is affected by the sequential position of subtasks within problems. Specifically, we segment all subtasks into five equal bins based on their relative positions and observe a consistent decline in model performance for subtasks appearing later in the sequence, as illustrated in Figure 12. This result is intuitive, as earlier subtasks typically impose stronger constraints on input variables, making them easier and prerequisites for subsequent subtasks. In contrast, later subtasks usually lack explicit constraints, requiring more generalized and optimized solutions.

# 5. In-Depth Analyses of Model Behavior and Error Patterns

We first analyze algorithmic complexity to identify models' strengths and weaknesses, next explore their strategic reasoning behaviors, then study the effect of contamination in our benchmark, investigate error patterns in model-

generated solutions, and finally examine programming-language effects.

## 5.1. Algorithmic Complexity Determines Model Performance Patterns

**Models are generally proficient at algorithm tags that require basic mathematical procedures and minimal compositional reasoning**. As shown in Table 3, all evaluated models consistently achieve higher pass rates on tasks categorized under implementation, mathematics, prefix sum, sorting, and graph traversal—GPT-5 notably attains over 70% accuracy on most of these tags. Such tasks primarily depend on leveraging similar concepts or procedural knowledge obtained from training. Performance noticeably declines for algorithms demanding deeper analytical reasoning, such as greedy methods and graph theory, where even top proprietary models like GPT-5 drop to around 60%. The greatest difficulties arise in tasks that require on-the-spot creative observations, intricate state designs, or hierarchical invariants—particularly evident in dynamic programming (DP), segment trees (ST), and tree (TR) problems, where GPT-5's pass rate sharply decreases to approximately 47%, 56%, and 38%, respectively. *To address these weaknesses*, future work could explore curriculum-driven fine-tuning (Huang et al., 2026) using carefully designed synthetic datasets of complex graph, tree, and DP problems, encouraging models to internalize the hierarchical invariants and compositional reasoning patterns crucial to solving these more challenging algorithmic tasks. Further, in Appendix E.6, we also find that increasing reasoning-token budgets yields greater improvements on complex tasks compared to parameter scaling or training strategy variations.

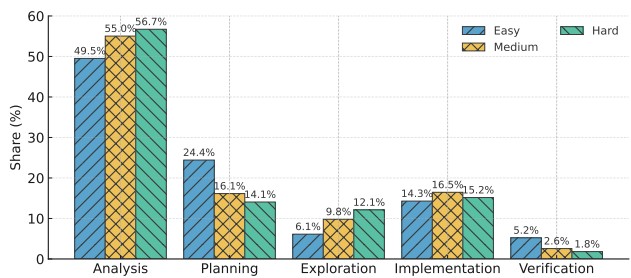

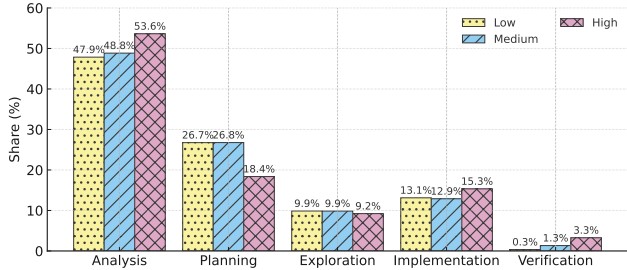

*(a)* GPT-OSS-120B-high across Easy/Medium/Hard. As problem difficulty increases, models prioritize exploration and analysis over planning and verification.

*(b)* GPT-OSS-120B across reasoning efforts. Higher reasoning budgets lead to deeper analysis, implementation, and verification without increased exploration.

*Figure 2.* **Reasoning Trace Analyses.** We categorize eight reasoning behaviors and divide them into five groups: **Analysis** (Algorithm/Proof analysis and Complexity Analysis), **Planning** (Problem Restatement and Subgoal Setting), **Exploration** (Backtracking and Dead-end recognition), **Implementation** (Pseudo implementation), **Verification** (Test Case Verification).

## 5.2. Reasoning Trace Analyses: Stronger Models Allocate Reasoning Tokens More Strategically

To better understand how thinking models solve challenging competitive coding problems, we conduct a detailed analysis on models' reasoning traces. Inspired by prior work (Gandhi et al., 2025; Ahmad et al., 2025) on reasoning behavior analysis, we categorize models' reasoning traces into eight behaviors and classify them into five groups as shown in Figure 2. Each trace is segmented into shorter chunks and annotated using GPT-OSS-120B. More details on the annotation prompt and implementation can be found in Appendix E.10.

**GPT-OSS-120B increases exploration and analysis with problem difficulty, yet maintains stable exploration levels across reasoning budgets.** In Figure 2a, on more challenging problems, GPT-OSS-120B-High devotes significantly more effort to exploration and deeper problem analysis, while it spends fewer tokens on initial planning and verification. This indicates that initial problem-structuring behaviors are typically conducted early and not revisited extensively once a potential solution path is identified. Notably, even when provided with increased reasoning budgets (from low to high effort), as shown in Figure 2b, GPT-OSS-120B strategically allocates extra tokens toward analysis, implementation, and verification, rather than further exploration. By maintaining stable exploration levels despite increased reasoning resources, the model mitigates excessive pivoting, a critical behavior that could lead to inefficient reasoning traces, or "underthinking" (Shojaee et al., 2025).

**Stronger reasoning models exhibit reduced exploration, allocating more resources toward solution development and analysis.** After controlling for problem difficulty and reasoning efforts, we further see in Figure 15a and Figure 19 that more capable models dedicate more reasoning tokens to problem understanding, structured planning, and detailed algorithmic analysis. Consequently, they spend less time pivoting to alternative paths, generating pseudo-

implementations, or performing test-case verification. This highlights a future direction: effectively allocating models' problem analysis and exploration to avoid excessive pivoting and prevent "underthinking".

**Initial planning behaviors and subsequent verification steps play crucial roles in models producing correct solutions.** Building upon this observation, we also investigate which reasoning behaviors distinguish correct from incorrect solutions. As illustrated in Figure 15b, correct solutions exhibit increased planning behaviors, potentially explaining why exploration behaviors diminish: well-structured planning facilitates clearly defined solution paths, reducing the need for exploratory detours. Additionally, correct solutions engage in verification behaviors more frequently, which slightly reduces the need for extensive analysis, as models are more confident in their reasoning. However, stronger models rely less on explicit verification overall due to robust upfront analysis and planning, which internalize many checks and reduce the need for post-hoc verification. Based on these insights, an important direction for future research is to optimize how models allocate reasoning effort across different cognitive behaviors.

## 5.3. Data Contamination Study

Following prior work (Riddell et al., 2024), we define contamination as artificially inflated and non-generalizing benchmark performance. In the context of code generation (Dekoninck et al., 2024), contamination may arise when models have encountered identical problems, official solutions, or structurally similar solution templates during training.

We evaluate contamination across four dimensions: temporal effects relative to training cutoffs, model familiarity with tasks or solutions, active statement perturbations, and code similarity to reference solutions.

**No evidence of temporal performance degradation.** In Figure 5 and Appendix E.1, we evaluate contamination risk

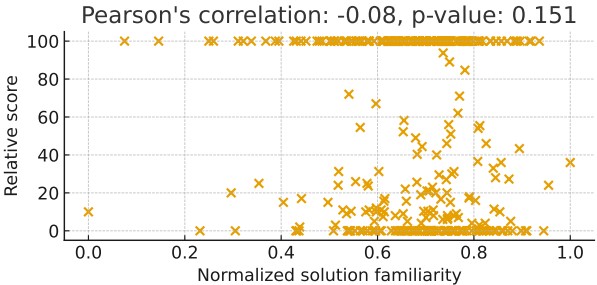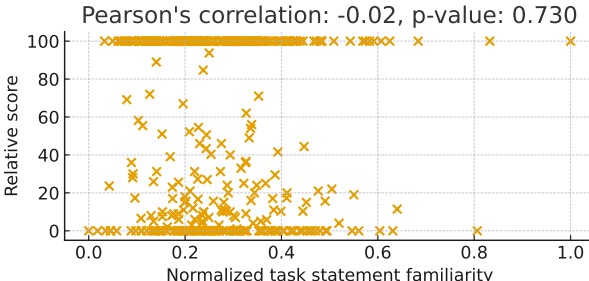

*Figure 3.* No significant positive correlation is observed between GPT-OSS-120B's familiarity with task statements and solutions (normalized via min-max scaling) and its performance, indicating that higher familiarity does not necessarily translate to better outcomes.

by examining whether model performance correlates with problem release dates, particularly around each model's training cutoff. Our analysis shows no meaningful relationship between publication time and performance: models do not perform better on older tasks, nor do they exhibit any noticeable drop in accuracy on problems released after their knowledge cutoff. This lack of temporal correlation suggests that broad data leakage is unlikely and provides evidence that our benchmark is not driven by memorization effects tied to publication dates.

**Familiarity with task statements and official solutions.** Following MLE-Bench (Chan et al., 2025) on detecting data contamination, we investigate whether the models' familiarity with task statements and official solutions affects their performance. Intuitively, if a model is more familiar with a task, it tends to perform better. Specifically, we define a model's familiarity with a document as the mean probability it assigns to each token, conditioned on all preceding tokens. Using GPT-OSS-120B, we compute this familiarity metric and plot it against pass rate (Figure 9) and relative score (Figure 3). Our analysis shows near-zero correlation between GPT-OSS-120B's familiarity with either task statements or official solutions and its performance, indicating minimal data contamination.

**Rephrasing preserves performance, while adversarial perturbations degrade it.** We further test whether models rely on memorized task statements by rewriting 71 problems with a strong rewriting model to change the surface form while preserving the algorithmic task. The rewritten variants keep the same input-output structure, constraints, and required solution logic, but replace narrative context, entity names, and incidental strings; Appendix E.4 shows an example. Rephrasing therefore separates semantic task understanding from surface-form recall: a model that understands the task should solve the original and rephrased statements similarly, whereas a model relying on memorized wording or problem identity should be more sensitive to the rewrite. As shown in Figure 4, GPT-OSS-120B, Seed-OSS, and Qwen3-30B-A3B maintain nearly unchanged relative scores under rephrasing, reducing the likelihood that performance is driven by direct statement memorization. In

contrast, the adversarial variants intentionally remove useful task cues: input/output examples are omitted, and random token masking makes the statement progressively less interpretable. Performance drops under these changes, especially at 30% and 50% masking, suggesting that models still depend on semantic task understanding rather than recovering solutions from memorized problem identities.

**Low similarity between models' solutions and official solutions.** Additionally, to investigate whether models rely primarily on memorization of template solutions rather than genuine reasoning, we employ the source code plagiarism detection tool Dolos (Maertens et al., 2024) to measure the similarity between accepted solutions generated by GPT-5 and Grok-4-Fast and the official solutions. This targets direct solution copying: if model outputs were retrieved from, or lightly edited from, official solutions, their source-code overlap would be high. GPT-5 and Grok-4-Fast have median similarities of 0.11 and 0.12, respectively; all scores remain below 0.50, and most are below 0.30 (Figure 10). We also test broader template leakage in Appendix E.5 and Appendix E.8: model-performance differences are nearly unrelated to official-solution similarity (GPT-5: $r = -0.084$, Grok-4-Fast: $r = -0.072$), and maximum similarity to CodeContest templates, computed from three filtered C++ solutions per seed problem, is only weakly correlated with relative score (GPT-5: $r = 0.136$, Grok-4-Fast: $r = 0.193$). These results do not rule out exposure to related algorithms, but they make direct solution-template copying unlikely to explain the observed benchmark performance.

### 5.4. Error Patterns in Model-Generated Code Submissions

**Stronger reasoning capabilities in models correlate with reduced failure rates, yet runtime errors remain a notable challenge.** In Appendix Figure 13, we analyze the submission status distribution across six selected models to better understand LLMs' solutions and their associated error patterns. As models exhibit stronger reasoning capabilities, their solutions show substantial reductions in failure types of time limit, memory limit, and compilation errors. However, runtime errors, although somewhat reduced, do not decline

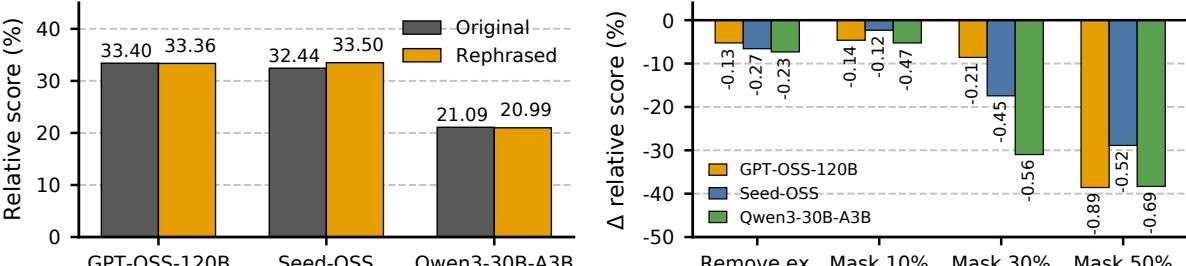

*Figure 4.* **Semantic rephrasing (Left):** semantically equivalent rephrasing preserves relative scores across three model families, reducing the likelihood that results are driven by direct statement memorization. **Adversarial perturbations (Right):** adversarial statement perturbations reduce relative scores, especially at higher token-masking ratios, suggesting that models depend on semantic task cues rather than memorized problem identities. Together, the two panels indicate that data contamination is unlikely to be the primary driver of the observed benchmark performance. Bar labels on the right show paired standardized effect sizes $d_z$, where negative values indicate degradation.

as sharply, highlighting persistent challenges in edge-case handling and execution robustness. We hypothesize that one possible reason top-performing models still exhibit relatively high runtime error rates could be their tendency to pursue more aggressive and optimized coding patterns, such as employing custom data structures, in-place transformations, and pointer arithmetic. These advanced techniques, while algorithmically sound, might inherently increase the potential for execution faults[6], especially in edge scenarios. Interestingly, GPT-OSS-20B displays compilation error rates comparable to weaker, non-reasoning-intensive models. We attribute this unexpected result to its cautious approach: the model often declines to generate solutions when it anticipates insufficient reasoning time, thereby triggering compilation-related failures. These findings highlight a limitation in the reinforcement learning approaches employed by current models (DeepSeek-AI et al., 2025; Yang et al., 2025a), which predominantly use solution correctness as the sole reward, neglecting efficiency and memory management. Future training techniques could incorporate fine-grained reward signals targeting these attributes, enabling models to optimize not only for correctness but also for reliable and efficient code execution.

### 5.5. Performance Variations Across Programming Languages

**C++ improves performance primarily through execution efficiency rather than different reasoning behavior.** To assess language-specific effects, we evaluate GPT-OSS-120B on 22 USACO contests covering 132 problems in Python, Java, and C++ (Appendix E.3). As shown in Table 14, C++ achieves the highest relative score and pass rate (53.26% and 59.09%), followed by Java (50.78% and 56.82%), while Python trails behind (44.35% and 46.97%). The perfor-

mance gap is most consistent with the execution constraints of competitive programming: C++ offers low overhead, predictable memory control, and optimized standard-library routines, while Python is more exposed to time limits and Java incurs runtime and memory-management overhead. However, the model's reasoning-behavior distribution remains similar across the three languages (Figure 7), suggesting that language choice changes implementation reliability and efficiency more than it changes the model's underlying problem-solving strategy.

## 6. Conclusion

In this work, we propose LiveOIBench, a comprehensive competitive coding benchmark featuring expert-curated OI-style tasks with detailed subtask rubrics, direct comparisons to human contestants, continuous updates with new Olympiad tasks to prevent contamination, and an offline evaluation system for accessible and reproducible assessment. We extensively evaluate 34 models including both proprietary and open-weight models. Our results highlight that proprietary models, particularly GPT-5, achieve impressive results but fall short of top human contestants. Among open-weight models, GPT-OSS, Seed-OSS, and Qwen3 demonstrate significant progress, with GPT-OSS-120B notably narrowing the gap to proprietary alternatives.

Further analyses reveal that current models particularly struggle with advanced algorithmic tasks, such as dynamic programming. Additionally, our reasoning trace analysis indicates that robust model performance relies on strategically allocating exploratory and analytical reasoning behaviors. Moving forward, we envision leveraging this benchmark to further investigate inference-time scaling strategies and training methods, particularly for challenging reasoning tasks. By offering a rigorous and reproducible benchmark, LiveOIBench aims to drive advancements in the reasoning and coding capabilities of LLMs.

---

[6]For instance, a simple algorithm like summing elements of an array becomes significantly more complex when highly optimized for memory access patterns using techniques such as loop unrolling and pragma directives in C++ (e.g., #pragma omp simd, #pragma unroll).

## Acknowledgments

We thank the anonymous reviewers for their valuable feedback and suggestions. We also thank the members of the LAUNCH Group at the University of Michigan for helpful discussions throughout this work. This research was supported in part through computational resources and services provided by Advanced Research Computing (ARC), a division of Information and Technology Services (ITS) at the University of Michigan, Ann Arbor. This work used the DeltaAI system at the National Center for Supercomputing Applications (NCSA) and the Bridges-2 system at the Pittsburgh Supercomputing Center (PSC). We also thank Amazon Web Services (AWS) for their generous support through AWS credits.

## Impact Statement

This work introduces a benchmark designed to evaluate the programming and algorithmic reasoning capabilities of frontier language models using problems derived from Informatics Olympiads. In particular, the benchmark is intended to measure model performance relative to highly skilled human contestants and to track progress toward, and potentially beyond, expert-level human performance on competitive programming tasks. We hope this benchmark will provide researchers and model developers with a rigorous and standardized framework for understanding the current limits of coding and reasoning systems. In constructing our benchmark, we carefully adhered to the licensing terms of all Informatics Olympiad materials, ensuring that data collection and redistribution complied with the policies of the original organizers.

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

id=YrycTjllL0.

# A. Dataset Construction

## A.1. Task Collection

We identified multiple Informatics Olympiad competitions and gathered all contests held from 2023 onward, along with their official website information. We specifically focused on the post-2022 period to minimize potential contamination from model training data. In total, we collected 72 contests, 50 of which include results from human contestants. The detailed statistics can be found in Table 7 and Table 8.

**Contest Information Extraction:** We developed a dedicated web crawler for each competition to extract task information directly from its official website. This includes task statements, test cases, reference and unofficial solutions, code attachments, time and memory limits, and detailed subtask specifications. We also parsed the contestant results pages and reformatted them into standardized CSV files. We passed the crawled contest result file into Gemini-2.5-Pro and prompted it to generate CSVs with normalized headers. Each file captures contestant names, countries, total and per-task scores, and awarded medals. We determined medal thresholds as follows. For general contests, the Gold, Silver, and Bronze thresholds are defined by the lowest total scores among participants who received each respective medal. For contests missing official medal data, we apply standard percentile cutoffs of 8–10% for Gold, 22–25% for Silver, and 50% for Bronze relative to the total number of participants. In contrast, for the USACO Bronze, Silver, and Gold contests, thresholds correspond to the minimum scores required to advance to the next competition level. In the USACO Platinum contest, medal assignments are based on the following score thresholds: a Gold medal is awarded to scores at or above the lowest top-ranked human score; scores above 350 receive a Silver medal; scores between 250 and 350 receive a Bronze medal; and scores of 250 or below receive no medal.

**Quality of Test Cases:** All test cases are sourced directly from official contest websites or reputable OJ platforms. These problems and their accompanying test suites are curated by organizing committees with extensive expertise in competitive programming. For example, tasks from the International Olympiad in Informatics (IOI) are designed by experts to meet rigorous specifications and undergo a multi-stage selection process to ensure originality and technical integrity. This construction process virtually eliminates false-positive solutions. Beyond validating crawled test cases with official ground-truth solutions, we manually inspect problems where all models achieve polarized scores (0 or 100) to ensure evaluation consistency.

**Markdown Conversion Quality Check:** We sampled 40 tasks (around 10% of task statements) while maintaining the same distribution across competition types as the full dataset. Our analysis revealed 4 tasks (10%) with conversion errors, primarily in nested table parsing within example inputs/outputs. While such formatting inconsistencies might affect human understanding, they have minimal effect on LLM comprehension because the core question semantics remain intact. GPT-5 fully solved three of the four tasks and passed over 70% of test cases on the remaining one, demonstrating that models are less sensitive to these formatting issues. An additional 14 tasks (35%) exhibited minor formatting issues, including header-level inconsistencies (e.g., "Input" formatted as # Input while "Output" is formatted as ### Output despite being at the same hierarchical level), incorrect superscript rendering, and uppercase/lowercase mismatches. These are inconsequential formatting artifacts that do not alter the semantic meaning of the problems. Our conversion pipeline uses Marker for PDF-to-markdown conversion followed by Gemini-2.0-Flash for automated verification and correction. After manually reviewing this 40-task sample, we confirmed that the parsed task statements are indeed of high quality, with only minor formatting errors that do not affect model performance. Confident in the robustness of these results, we proceeded with batch processing the remaining tasks. We believe these slight formatting inconsistencies are inconsequential and do not compromise the validity of our evaluation.

**Missing Data:** When the official website lacks complete or up-to-date contest information, we enhance our dataset by retrieving the missing details from reputable secondary platforms such as CSES and LibreOJ. These platforms host curated repositories of contest materials and metadata, and contain a substantial number of user submissions along with their corresponding pass rates. Their widespread adoption within the competitive programming and informatics communities suggests high accuracy and reliability. For contests missing test cases on the official site, we employ a parser to retrieve them from CSES and integrate them into our dataset. If official code solutions are absent or invalid, we obtain five user-submitted solutions from LibreOJ that achieved a 100% pass rate and include them in the dataset. Valid solutions from open-source GitHub repositories are also downloaded to enhance the dataset. By supplementing incomplete primary data with these established sources, we ensure our dataset maintains high standards of accuracy and completeness.

## A.2. Problem Filtering and Solution Verification

To ensure that the solutions collected from official websites and external platforms are accurate, we create an evaluation code judge to validate whether our solution can pass all the test cases from our dataset. The code judge operates differently based on the question type. If a question is removed from a contest, we exclude it from the analysis when comparing model scores against human performance.

**Batch:**  For all the batch problems, we run the official code solution against the input-output test cases. The input file is provided to the program, and the code output is verified against the expected output. The subtask scores are computed to verify that the total score adds up to the total points. Any problem for which the solution failed a test case or produced an invalid total score was excluded from further analysis. For problems that accept multiple valid outputs, we set up a testing environment using the grader file supplied in the contest materials and apply the same evaluation procedure, disregarding problems with incorrect solution files.

**Interactive:**  If the problem type is interactive, the grader file is executed first to establish the testing environment. Subsequently, the solution file is launched within the environment to exchange input/output streams interactively. After the problem finishes, the grader's evaluation output is collected to determine whether the solution passed. If the grader does not return a full mark on the ground-truth solution, the corresponding problem is discarded.

**Output-Only:**  We exclude output-only problems since they do not require contestants to submit algorithmic code solutions, making them difficult to use for evaluating model performance.

## A.3. Metadata Collection

To further enrich and structure our dataset, we augment it with comprehensive problem metadata crawled from solved.ac and Luogu , capturing difficulty ratings and algorithm tags. We then utilize Gemini-2.0-Flash to semantically match problems across different platforms, resolving inconsistencies in label formats and taxonomies through a unified mapping strategy.

**Difficulty Tags:**  Solved.ac uses integer values from 1 to 30 to represent the difficulty levels, where 1 corresponds to the easiest tier (Bronze V) and 30 corresponds to the hardest tier (Ruby I). However, Luogu employs 7 categorical text labels for its difficulty. To reconcile the inconsistent difficulty scales across platforms, we construct a numerical mapping on a $0 - 30$ scale for Luogu, translating the native difficulty descriptor tags into standardized numerical scores using the mapping as specified in Table 4. The unified scale enables us to assign difficulty scores to all problems by taking the union of both sources.

*Table 4.* Difficulty Tags and Corresponding Scores

| Difficulty Tag | Difficulty Score |
|:---:|:---:|
| Beginner | 5 |
| Easy | 9 |
| Intermediate | 13 |
| Hard | 16 |
| Advanced | 18 |
| Expert | 21 |
| Master | 24 |

**Algorithm Tags.** To ensure data integrity and consistency, we develop a normalization dictionary to standardize dataset labels. This dictionary systematically resolves lexical and semantic variations, including synonyms, related terms, and differences in granularity, by mapping them to a unified set of canonical tags.

**Missing Tags.** In cases where tags were missing, we utilize Gemini-2.0-Flash to infer plausible labels and difficulty from the problem description, enhancing both completeness and labeling quality. To assess the reliability of LLM-inferred difficulty scores, we conduct sampling-based validation on problems with existing difficulty annotations and observe a high degree of consistency with their original scores.

**Divisions.** Finally, we analyze the distribution of algorithm and difficulty tags across the corpus and partition the difficulty range of all contests into four divisions, thereby improving robustness and facilitating downstream contest categorization. The division boundaries are listed in Table 5.

**Problem Difficulty.** We sort all task difficulty scores and split them into three equal-sized buckets by taking the empirical

*Table 5.* Division Boundaries by Difficulty

| Division | Min Difficulty | Max Difficulty | Avg Difficulty | Total Contests |
|----------|---------------|----------------|----------------|----------------|
| Division 4 | 5.0 | 15.78 | 13.76 | 17 |
| Division 3 | 16.0 | 20.33 | 18.05 | 19 |
| Division 2 | 20.33 | 22.33 | 21.52 | 19 |
| Division 1 | 22.5 | 30.0 | 23.62 | 17 |

one-third and two-thirds cut points. The problem difficulty distribution is listed in Table 6

*Table 6.* Problem difficulty distribution using quantile thresholds.

| Level | # Problems | % of Total | Threshold Rule |
|-------|-----------|------------|----------------|
| Easy | 143 | 35.48% | $d \leq 17$ |
| Medium | 144 | 35.73% | $18 \leq d \leq 22$ |
| Hard | 116 | 28.78% | $d \geq 23$ |
| **Total** | **403** | **100%** | – |

### A.4. Codeforces Ratings Collection

**Result Collection:** For each contest, the raw human results files are downloaded and restructured. These files typically include contestant identifiers such as usernames, countries, individual task scores, total scores, and medal information.

**Rating Data Retrieval:** Codeforces rating data are obtained by algorithmically mapping contestants' names to their corresponding profiles in the Codeforces database. Usernames are first normalized by removing diacritics and converting all text to lowercase to enhance matching robustness. Using these normalized usernames together with each contestant's country, our program submits Google Search queries and inspects the top results to identify potential Codeforces profile URLs. When valid Codeforces URLs are identified, the extracted handles are queried via the Codeforces API to obtain detailed user profile information, including full name, country, Codeforces ID, and rating history.

**Database Generation:** The retrieved rating histories are parsed to extract Codeforces ratings for each year from 2022 to 2025. When annual data are unavailable, we backfill missing entries by using contestants' most recent available Codeforces ratings from prior years. For instance, if a contestant participates in 2025 but lacks an updated rating, we use their previous rating when applicable. Contest names and contestant metadata are then appended to a master Codeforces database. If a contestant's record already exists, only new contest information is added to the existing profile.

**Rating Matching:** After these procedures are applied to every contest results file, a database of Codeforces ratings for all contestants is established. Finally, we link each contestant's Codeforces rating—matched by name and country—to the corresponding contest year.

**Model Ratings:** To benchmark model performance on our dataset, we calculate a corresponding Codeforces rating for each model on every contest. For each task, we present the full problem statement to the models and prompt them to generate code solutions. Using the provided subtask and test-case data, we compute the total score of each model's solution. After obtaining total scores, we derive Codeforces ratings for the models using the CodeElo formula (Quan et al., 2025) given below, where $m$ is the expected rank of a contestant (or model) with rating $r$, compared to $n$ contestants with known Codeforces ratings $r_{(i)}$.

$$m = \sum_{i=1}^{n} \frac{1}{1 + 10^{\frac{r - r_{(i)}}{400}}}$$

To ensure the reliability and accuracy of our analysis, we perform several filtering steps on the human data prior to computing Elo ratings. First, we exclude participants who either lack an official Codeforces rating or whose ratings fall below 500. Next, we identify and remove performance outliers by fitting a third-degree polynomial regression to the score-rating data and discarding any results lying more than 2 standard deviations from the fitted curve.

Finally, to reduce statistical noise and further enhance data quality, we exclude contests with fewer than 15 valid human Codeforces ratings from the Elo calculation. These steps collectively ensure that our resulting model ratings reliably reflect the true relationship between contestants' Codeforces ratings and their total contest scores. Table 9 presents all contests for

which we successfully matched contestants to their Codeforces profiles, along with the median Codeforces rating for each contest.

**Codeforces Profile Quality Check:** To verify the accuracy of matching contestants to their Codeforces profiles, we performed a detailed manual error analysis. We randomly sampled five contest results, selecting ten contestants from each contest. Using the same search query (contestant name, country, and "Codeforces"), we located each contestant's Codeforces profile and verified the match by comparing the name, country, username, and yearly rating history against our stored records. Our analysis confirmed that all 50 contestant profiles matched perfectly, demonstrating 100% accuracy and confirming the high precision of our profile-matching pipeline. Nevertheless, we recognize that some contestants may lack Codeforces profiles or use alternative names on their profiles, potentially causing missed matches. Given the high precision of our pipeline, the individualized nature of Elo rating computations, and our exclusion of contests containing fewer than 15 validated contestant profiles, we anticipate that such omissions will minimally affect the overall accuracy of the model's rating estimates.

### A.5. Benchmark Maintenance

We plan to update LiveOIBench every 3–6 months, adjusting our schedule to incorporate newly released models and the latest contest problems as they become available. We prioritize the inclusion of contests that offer sufficient difficulty to challenge strong reasoning models. Our maintenance workflow is highly automated: once a new contest URL is identified, a custom crawler retrieves the problem data. PDF statements and contestant results are then parsed via an LLM-based extraction pipeline. To ensure data quality, we validate the crawled test cases by executing official ground-truth solutions. Metadata, including algorithmic tags, difficulty labels, and contestant Codeforces ratings, are collected via dedicated crawlers for Luogu, Solved.AC, and Codeforces. This pipeline requires minimal human intervention beyond final verification of crawled results.

Finally, we run evaluation using our evaluation framework and update the benchmark leaderboard. When integrating a new model, we evaluate it against all contests after its knowledge cutoff date or its release date if the cutoff date is not available; conversely, new contests are evaluated against models released within the most recent year to ensure relevance. Users can leverage a time-filter feature on our leaderboard for performance analysis. All code is available in our repository.

*Table 7.* Statistics of different competitions. USACO does not provide subtask information.

| Competitions | Total Tasks | Total Contests | Avg. Subtasks | Test Cases/Task | Token Count | Difficulty |
|---|---|---|---|---|---|---|
| IOI | 12 | 2 | 7.08 | 112.42 | 2359.58 | 22.83 |
| BOI | 18 | 3 | 6.22 | 110.83 | 1139.72 | 22.28 |
| CEOI | 11 | 2 | 7.45 | 89.45 | 1339.36 | 22.33 |
| EGOI | 13 | 2 | 5.31 | 87.23 | 1388.85 | 18.50 |
| EJOI | 12 | 2 | 7.25 | 54.92 | 1443.08 | 12.00 |
| IATI | 11 | 2 | 6.82 | 78.09 | 1302.00 | 23.03 |
| OOI | 32 | 4 | 8.88 | 128.19 | 1639.31 | 23.02 |
| RMI | 12 | 2 | 6.33 | 37.42 | 896.42 | 23.00 |
| APIO | 5 | 2 | 8.00 | 58.80 | 2052.40 | 21.67 |
| JOI | 42 | 7 | 5.79 | 103.00 | 1848.29 | 21.17 |
| CCO | 32 | 6 | 4.19 | 63.34 | 754.25 | 13.36 |
| COCI | 62 | 13 | 3.69 | 55.02 | 897.05 | 16.38 |
| NOI | 9 | 3 | 6.11 | 63.22 | 970.00 | 21.89 |
| USACO | 132 | 22 | - | 17.11 | 751.07 | 19.13 |
| Division 1 | 87 | 17 | 7.42 | 85.45 | 1440.46 | 23.62 |
| Division 2 | 89 | 19 | 5.23 | 80.16 | 1288.83 | 21.52 |
| Division 3 | 115 | 19 | 7.32 | 57.85 | 1124.07 | 18.05 |
| Division 4 | 112 | 17 | 3.80 | 28.54 | 738.32 | 13.76 |
| All Competitions | 403 | 72 | 5.80 | 60.59 | 1121.55 | 19.04 |

*Table 8.* Contest dates from 2023–2025 for major Olympiads.

| Contest | Date | Human Results |
|---|---|---|
| Asia-Pacific Informatics Olympiad 2023 | 2023-05-20 | True |
| Asia-Pacific Informatics Olympiad 2024 | 2024-05-18 | True |
| Baltic Olympiad in Informatics 2023 | 2023-04-28 | True |
| Baltic Olympiad in Informatics 2024 | 2024-05-03 | True |
| Baltic Olympiad in Informatics 2025 | 2025-04-29 | True |
| Canadian Computing Olympiad 2023 CCC_Junior | 2023-02-15 | True |
| Canadian Computing Olympiad 2023 CCC_Senior | 2023-02-15 | True |
| Canadian Computing Olympiad 2023 CCO | 2023-05-29 | True |
| Canadian Computing Olympiad 2024 CCC_Junior | 2024-02-21 | True |
| Canadian Computing Olympiad 2024 CCC_Senior | 2024-02-27 | True |
| Canadian Computing Olympiad 2024 CCO | 2024-05-27 | True |
| Central European Olympiad in Informatics 2023 | 2023-08-13 | True |
| Central European Olympiad in Informatics 2024 | 2024-06-24 | True |
| Croatian Open Competition in Informatics 2023 CONTEST_#3 | 2023-01-14 | True |
| Croatian Open Competition in Informatics 2023 CONTEST_#4 | 2023-02-11 | True |
| Croatian Open Competition in Informatics 2023 CONTEST_#5 | 2023-03-11 | True |
| Croatian Open Competition in Informatics 2024 CONTEST_#1 | 2023-11-04 | True |
| Croatian Open Competition in Informatics 2024 CONTEST_#2 | 2023-12-02 | True |
| Croatian Open Competition in Informatics 2024 CONTEST_#3 | 2024-01-13 | True |
| Croatian Open Competition in Informatics 2024 CONTEST_#4 | 2024-02-10 | True |
| Croatian Open Competition in Informatics 2024 CONTEST_#5 | 2024-03-16 | True |
| Croatian Open Competition in Informatics 2025 CONTEST_#1 | 2024-10-05 | True |
| Croatian Open Competition in Informatics 2025 CONTEST_#2 | 2024-11-09 | True |
| Croatian Open Competition in Informatics 2025 CONTEST_#3 | 2024-12-07 | True |
| Croatian Open Competition in Informatics 2025 CONTEST_#4 | 2025-01-25 | True |
| Croatian Open Competition in Informatics 2025 CONTEST_#5 | 2025-02-15 | True |
| European Girls' Olympiad in Informatics 2023 | 2023-07-15 | True |
| European Girls' Olympiad in Informatics 2024 | 2024-07-21 | True |
| European Junior Olympiad in Informatics 2023 | 2023-09-08 | True |
| European Junior Olympiad in Informatics 2024 | 2024-08-16 | True |
| International Advanced Tournament in Informatics 2024 junior | 2024-04-17 | True |
| International Advanced Tournament in Informatics 2024 senior | 2024-04-17 | True |
| International Olympiad in Informatics 2023 | 2023-08-28 | True |
| International Olympiad in Informatics 2024 | 2024-09-01 | True |
| Japanese Olympiad in Informatics 2023 JOI | 2023-02-12 | True |
| Japanese Olympiad in Informatics 2023 JOI_open | 2023-08-05 | True |
| Japanese Olympiad in Informatics 2023 JOI_spring | 2023-03-19 | True |
| Japanese Olympiad in Informatics 2024 JOI | 2024-02-04 | True |
| Japanese Olympiad in Informatics 2024 JOI_open | 2024-06-17 | True |
| Japanese Olympiad in Informatics 2024 JOI_spring | 2024-03-21 | True |
| Japanese Olympiad in Informatics 2025 JOI | 2025-02-02 | True |
| Nordic Olympiad in Informatics 2023 | 2023-03-22 | True |
| Nordic Olympiad in Informatics 2024 | 2024-03-06 | True |
| Nordic Olympiad in Informatics 2025 | 2025-03-05 | True |
| Open Olympiad in Informatics 2023 final | 2024-03-07 | True |
| Open Olympiad in Informatics 2023 qualification | 2023-11-25 | True |
| Open Olympiad in Informatics 2024 final | 2025-03-06 | True |
| Open Olympiad in Informatics 2024 qualification | 2024-12-01 | True |

| Contest | Date | Human Results |
|---|---|---|
| Romanian Master of Informatics 2023 | 2023-10-11 | True |
| Romanian Master of Informatics 2024 | 2024-11-27 | True |
| USA Computing Olympiad 2023 December_Contest-combined | 2022-12-15 | False |
| USA Computing Olympiad 2023 December_Contest-platinum | 2022-12-15 | False |
| USA Computing Olympiad 2023 February_Contest-combined | 2023-02-24 | False |
| USA Computing Olympiad 2023 February_Contest-platinum | 2023-02-24 | False |
| USA Computing Olympiad 2023 January_Contest-combined | 2023-01-27 | False |
| USA Computing Olympiad 2023 January_Contest-platinum | 2023-01-27 | False |
| USA Computing Olympiad 2023 US_Open_Contest-combined | 2023-03-24 | False |
| USA Computing Olympiad 2023 US_Open_Contest-platinum | 2023-03-24 | False |
| USA Computing Olympiad 2024 December_Contest-combined | 2023-12-13 | False |
| USA Computing Olympiad 2024 December_Contest-platinum | 2023-12-13 | False |
| USA Computing Olympiad 2024 February_Contest-combined | 2024-02-16 | False |
| USA Computing Olympiad 2024 February_Contest-platinum | 2024-02-16 | False |
| USA Computing Olympiad 2024 January_Contest-combined | 2024-01-26 | False |
| USA Computing Olympiad 2024 January_Contest-platinum | 2024-01-26 | False |
| USA Computing Olympiad 2024 US_Open_Contest-combined | 2024-03-15 | False |
| USA Computing Olympiad 2024 US_Open_Contest-platinum | 2024-03-15 | False |
| USA Computing Olympiad 2025 February_Contest-combined | 2025-02-21 | False |
| USA Computing Olympiad 2025 February_Contest-platinum | 2025-02-21 | False |
| USA Computing Olympiad 2025 January_Contest-combined | 2025-01-24 | False |
| USA Computing Olympiad 2025 January_Contest-platinum | 2025-01-24 | False |
| USA Computing Olympiad 2025 US_Open_Contest-combined | 2025-03-21 | False |
| USA Computing Olympiad 2025 US_Open_Contest-platinum | 2025-03-21 | False |
| Total: 72 | | 50 |

*Table 9.* Summary of Human Codeforces ratings for various contests.

| Contest | Contestants | Median Rating |
|---|---|---|
| Asia-Pacific Informatics Olympiad 2023 | 60 | 2184.85 |
| Asia-Pacific Informatics Olympiad 2024 | 72 | 2108.28 |
| Baltic Olympiad in Informatics 2023 | 24 | 2006.12 |
| Baltic Olympiad in Informatics 2024 | 27 | 1973.11 |
| Baltic Olympiad in Informatics 2025 | 19 | 2023.37 |
| Canadian Computing Olympiad 2023 CCC_Junior | 185 | 1993.04 |
| Canadian Computing Olympiad 2023 CCC_Senior | 88 | 2141.22 |
| Canadian Computing Olympiad 2023 CCO | 7 | 2379.14 |
| Canadian Computing Olympiad 2024 CCC_Junior | 228 | 1822.74 |
| Canadian Computing Olympiad 2024 CCC_Senior | 98 | 1960.28 |
| Central European Olympiad in Informatics 2023 | 28 | 2214.57 |
| Central European Olympiad in Informatics 2024 | 27 | 2156.81 |
| Croatian Open Competition in Informatics 2023 CONTEST_#3 | 10 | 2050.7 |
| Croatian Open Competition in Informatics 2023 CONTEST_#4 | 10 | 2050.7 |
| Croatian Open Competition in Informatics 2023 CONTEST_#5 | 10 | 2050.7 |
| Croatian Open Competition in Informatics 2024 CONTEST_#1 | 65 | 1795.92 |
| Croatian Open Competition in Informatics 2024 CONTEST_#2 | 55 | 1807.35 |
| Croatian Open Competition in Informatics 2024 CONTEST_#3 | 61 | 1873.16 |
| Croatian Open Competition in Informatics 2024 CONTEST_#4 | 55 | 1756.38 |
| Croatian Open Competition in Informatics 2024 CONTEST_#5 | 58 | 1744.55 |
| Croatian Open Competition in Informatics 2025 CONTEST_#1 | 5 | 2016.6 |
| Croatian Open Competition in Informatics 2025 CONTEST_#2 | 5 | 2016.6 |
| Croatian Open Competition in Informatics 2025 CONTEST_#3 | 5 | 2016.6 |
| Croatian Open Competition in Informatics 2025 CONTEST_#4 | 5 | 2016.6 |
| Croatian Open Competition in Informatics 2025 CONTEST_#5 | 5 | 2016.6 |
| European Girls' Olympiad in Informatics 2023 | 54 | 1646.02 |
| European Girls' Olympiad in Informatics 2024 | 31 | 1678.23 |
| European Junior Olympiad in Informatics 2023 | 22 | 1876.0 |
| European Junior Olympiad in Informatics 2024 | 32 | 1877.16 |
| International Olympiad in Informatics 2023 | 216 | 2105.12 |
| International Olympiad in Informatics 2024 | 253 | 2115.76 |
| Japanese Olympiad in Informatics 2023 JOI | 139 | 2314.65 |
| Japanese Olympiad in Informatics 2023 JOI_open | 98 | 2195.65 |
| Japanese Olympiad in Informatics 2023 JOI_spring | 252 | 2278.29 |
| Japanese Olympiad in Informatics 2024 JOI | 144 | 2022.38 |
| Japanese Olympiad in Informatics 2024 JOI_open | 102 | 2263.97 |
| Japanese Olympiad in Informatics 2024 JOI_spring | 245 | 2221.79 |
| Nordic Olympiad in Informatics 2023 | 16 | 1695.5 |
| Nordic Olympiad in Informatics 2024 | 13 | 1726.08 |
| Nordic Olympiad in Informatics 2025 | 6 | 1687.67 |
| Open Olympiad in Informatics 2023 final | 142 | 2028.51 |
| Open Olympiad in Informatics 2023 qualification | 92 | 1421.75 |
| Open Olympiad in Informatics 2024 final | 69 | 2037.86 |
| Open Olympiad in Informatics 2024 qualification | 87 | 1512.4 |
| Romanian Master of Informatics 2023 | 75 | 1953.19 |
| Romanian Master of Informatics 2024 | 93 | 1970.59 |

**A.6. Sample Task**

We now present an example drawn from the *International Olympiad in Informatics 2024*. The following task, titled *Nile*, illustrates a typical problem style in our dataset.

---

**Problem: Nile**

You want to transport $N$ artifacts through the Nile. The artifacts are numbered from 0 to $N - 1$. The weight of artifact $i$ ($0 \le i < N$) is $W[i]$.
To transport the artifacts, you use specialized boats. Each boat can carry **at most two** artifacts.

- If you decide to put a single artifact in a boat, the artifact weight can be arbitrary.

- If you want to put two artifacts in the same boat, you have to make sure the boat is balanced evenly. Specifically, you can send artifacts $p$ and $q$ ($0 \le p < q < N$) in the same boat only if the absolute difference between their weights is at most $D$, i.e. $|W[p] - W[q]| \le D$.

The cost of transporting artifact $i$ ($0 \le i < N$) is:

- $A[i]$, if you put the artifact in its own boat, or

- $B[i]$, if you put it in a boat together with some other artifact.

If artifacts $p$ and $q$ are sent together, the total cost is $B[p] + B[q]$. Since $B[i] < A[i]$ for all $i$, sending an artifact with another is always cheaper when possible.
Unfortunately, the river is unpredictable and the value of $D$ changes often. Your task is to answer $Q$ queries, described by array $E$ of length $Q$. For query $j$ ($0 \le j < Q$), the answer is the minimum cost of transporting all $N$ artifacts when $D = E[j]$.

**Implementation Details**

```
std::vector<long long> calculate_costs(
    std::vector<int> W,
    std::vector<int> A,
    std::vector<int> B,
    std::vector<int> E)
```

- `W`, `A`, `B`: arrays of length $N$, describing weights and costs.

- `E`: array of length $Q$, values of $D$.

- Returns: array $R$ with $R[j]$ equal to the minimum cost for $D = E[j]$.

**Constraints**

$1 \le N \le 100,000$
$1 \le Q \le 100,000$
$1 \le W[i] \le 10^9$ for each $i$ such that $0 \le i < N$
$1 \le B[i] < A[i] \le 10^9$ for each $i$ such that $0 \le i < N$
$1 \le E[j] \le 10^9$ for each $j$ such that $0 \le j < Q$

**Subtasks**

| Subtask | Score | Additional Constraints |
|---------|-------|------------------------|
| 1 | 6 | $Q \le 5$; $N \le 2000$; $W[i] = 1$ for each $i$ such that $0 \le i < N$ |
| 2 | 13 | $Q \le 5$; $W[i] = i + 1$ for each $i$ such that $0 \le i < N$ |
| 3 | 17 | $Q \le 5$; $A[i] = 2$ and $B[i] = 1$ for each $i$ such that $0 \le i < N$ |
| 4 | 11 | $Q \le 5$; $N \le 2000$ |
| 5 | 20 | $Q \le 5$ |
| 6 | 15 | $A[i] = 2$ and $B[i] = 1$ for each $i$ such that $0 \le i < N$ |
| 7 | 18 | No additional constraints. |

**Example**

```
calculate_costs([15, 12, 2, 10, 21],
                [5, 4, 5, 6, 3],
                [1, 2, 2, 3, 2],
                [5, 9, 1]) -> [16, 11, 23]
```

Explanation:

- $D = 5$: pair $(0, 3)$, others alone $\Rightarrow 16$

- $D = 9$: pairs $(0, 1)$ and $(2, 3)$, artifact 4 alone $\Rightarrow 11$

- $D = 1$: no pairs possible, all alone $\Rightarrow 23$

**Sample Grader**

*Input format:*

```
N
W[0] A[0] B[0]
W[1] A[1] B[1]
...
W[N-1] A[N-1] B[N-1]
Q
E[0]
E[1]
...
E[Q-1]
```

*Output format:*

```
R[0]
R[1]
...
R[S-1]
```

where $S = Q$ is the length of the output array.

---

### grader.cpp

```cpp
#include "nile.h"
#include <cstdio>
#include <vector>

int main() {
  int N; scanf("%d", &N);
  std::vector<int> W(N), A(N), B(N);
  for (int i = 0; i < N; i++)
    scanf("%d%d%d", &W[i], &A[i], &B[i]);
  int Q; scanf("%d", &Q);
  std::vector<int> E(Q);
  for (int j = 0; j < Q; j++)
    scanf("%d", &E[j]);

  auto R = calculate_costs(W, A, B, E);
  for (auto x : R) printf("%lld\n", x);
}
```

**Problem Metadata**

```
"nile": {
        "id": 32266,
        "title": "Nile",
        "difficulty": 19,
        "tags": ["data structures", "segment tree",
        "disjoint set", "offline queries"],
        "time_limit": 2.0,
        "memory_limit": 2048.0,
        "task_type": "Batch"
        }
```

## A.7. Competition Information

**International Olympiad in Informatics (IOI)**   First held in 1989, the IOI is the annual world championship for informatics. Participants are organized into national delegations, with each of the approximately 90 participating countries sending a team of up to four students. These contestants are selected through highly rigorous, multi-stage national olympiads.

**Baltic Olympiad in Informatics (BOI)**   Established in 1995, the BOI brings together teams from countries bordering the Baltic Sea and invited guest nations. Each member country's national informatics organization selects a team of their top-ranking secondary school students, who are often candidates for that year's IOI team.

**Central European Olympiad in Informatics (CEOI)**   Originating in 1994, the CEOI is an on-site competition for teams from Central European member countries and several guest nations. Delegations are chosen by respective national olympiad committees and are typically composed of students who have achieved top results in their national contests.

**European Girls' Olympiad in Informatics (EGOI)**   An initiative from 2021, the EGOI is an international competition for teams from European and guest countries. Each participating country selects a team of up to four female secondary school students who have demonstrated strong performance in their national-level informatics competitions.

**European Junior Olympiad in Informatics (EJOI)**   Founded in 2017, the EJOI is a major international event for a younger age group. Each European member country sends a national delegation of up to four students who are under the age of 15.5. Participants are typically the winners of national junior-level informatics olympiads.

**International Advanced Tournament in Informatics (IATI)**   Established in 2009 and hosted in Shumen, Bulgaria, the IATI is an international competition with two distinct age divisions, Junior and Senior. It brings together national and regional teams from numerous participating countries. Contestants are typically selected by their national informatics organizations based on strong results in previous competitions.

**Open Olympiad in Informatics (OOI)**   The Open Olympiad in Informatics (OOI) is the final stage of the All-Russian Olympiad in Informatics. Its participants are composed of two groups: the top Russian students who have advanced through a rigorous nationwide selection process, and official teams from various guest countries that receive a formal invitation to compete.

**Romanian Master of Informatics (RMI)**   First held in 2009, the RMI is a prestigious international competition. Participation is by invitation only; the organizers invite official national teams from countries with a strong track record at the IOI. This makes the participant pool one of the strongest in the world.

**Asia-Pacific Informatics Olympiad (APIO)**   The APIO, an online contest since 2007, involves students from countries and regions across the Asia-Pacific. Each member region organizes its own contest to select a set of national participants, who then compete from a supervised site within their home country.

**Japanese Olympiad in Informatics (JOI)**   Since 1994, the JOI has served as Japan's national selection process. It is open to Japanese junior high and high school students, who compete in preliminary rounds. Top performers are then invited to an exclusive on-site final and training camp, from which the IOI team is chosen.

**Canadian Computing Olympiad (CCO)**   The CCO, since 1996, is the invitational final stage of Canada's national selection process. Participation is granted to the top $20 - 25$ senior-level students from the open Canadian Computing Competition (CCC), who then compete to form the four-member IOI team.

**Croatian Open Competition in Informatics (COCI)**   Since 2006, COCI has operated as an online contest series open to individual participants worldwide. For Croatian students, cumulative performance across the year's rounds is a primary component in the selection process for the national team for the IOI and other international events.

**Nordic Olympiad in Informatics (NOI)** The Nordic Olympiad in Informatics brings together top secondary school students from Denmark, Finland, Iceland, Norway, and Sweden. Each country selects its participants based on the results of their respective national olympiads, with the NOI serving as a key qualifier for the BOI.

**USA Computing Olympiad (USACO)** The USACO is an open competition primarily for pre-college students in the United States, though it attracts many international participants. Its monthly online contests determine which top US-based students in the Platinum division are invited to a training camp, where the four-member IOI team is selected.

## B. Model Information

- **Proprietary LLMs:** This category includes high-performing proprietary models such as Gemini-2.5 (Comanici et al., 2025), GPT-o3-Mini-High (OpenAI, 2025c), and GPT-4.1 (OpenAI, 2025a).

- **Open-weight Thinking LLMs:** These are openly available models that are empowered with inherent thinking or reasoning capabilities. This group includes Qwen3 (Yang et al., 2025a) and DeepSeek-R1 (DeepSeek-AI et al., 2025), as well as those distilled from DeepSeek-R1.

- **Open-weight Non-Thinking LLMs:** This category consists of openly available models that are not equipped with intrinsic thinking mechanisms. This includes DeepSeek Coder-V2 (DeepSeek-AI et al., 2024b), DeepSeek-V3 (DeepSeek-AI et al., 2024a), Qwen2.5 (Yang et al., 2024), Qwen2.5-Coder (Hui et al., 2024), Qwen3 (Yang et al., 2025b), Mistral (Jiang et al., 2023) and Llama-3 (Dubey et al., 2024).

- Refer to Table 10 and Table 11 for more details. We use the default parameter setting if it's not listed.

- We use the prompt in Table B for all models.

*Table 10.* Model list of Non-Thinking LLMs with model providers and decoding settings

| Non-Thinking LLMs | Model Provider | Temperature | Top-p |
|---|---|---|---|
| GPT-4.1 (OpenAI, 2025a) | OpenAI | 1.0 | 1.0 |
| Qwen2.5-72B-Instruct (Yang et al., 2024) | Alibaba | 0.7 | 0.8 |
| Qwen2.5-Coder-32B-Instruct (Hui et al., 2024) | Alibaba | 0.7 | 0.8 |
| Qwen2.5-Coder-14B-Instruct (Hui et al., 2024) | Alibaba | 0.7 | 0.8 |
| Qwen2.5-Coder-7B-Instruct (Hui et al., 2024) | Alibaba | 0.7 | 0.8 |
| Mistral-Large-Instruct-2411 (Jiang et al., 2023) | Mistral | 1.0 | 1.0 |
| Mistral-Small-3.1-24B-2503 (Jiang et al., 2023) | Mistral | 1.0 | 1.0 |
| LLaMa-4-Scout (Meta AI, 2025) | Meta | 0.6 | 0.9 |
| LLaMa-3.3-70B-Instruct (Dubey et al., 2024) | Meta | 0.6 | 0.9 |
| LLaMa-3.1-8B-Instruct (Dubey et al., 2024) | Meta | 0.6 | 0.9 |
| DeepSeek-V3 (DeepSeek-AI et al., 2024a) | DeepSeek | 1.0 | 1.0 |
| DeepSeek-Coder-V2-Lite-Instruct (DeepSeek-AI et al., 2024b) | DeepSeek | 0.3 | 0.95 |
| Codestral-22B-v0.1 (Mistral AI team, 2024) | Mistral | 0.0 | 1.0 |

*Table 11.* Model list with categories, including model names, organizations, reasoning budget, and decoding settings.

| Thinking LLMs | Model Provider | Reasoning Budget (Max Tokens) | Temperature | Top-p |
|---|---|---|---|---|
| GPT-5 (OpenAI, 2025b) | OpenAI | Medium (100K) | - | - |
| GPT-O3-Mini-High (OpenAI, 2025c) | OpenAI | High (100K) | - | - |
| GPT-OSS-120B-High (OpenAI et al., 2025) | OpenAI | High (128K) | 1.0 | 1.0 |
| GPT-OSS-20B-High (OpenAI et al., 2025) | OpenAI | High (128K) | 1.0 | 1.0 |
| GPT-OSS-120B (OpenAI et al., 2025) | OpenAI | Medium (128K) | 1.0 | 1.0 |
| GPT-OSS-20B (OpenAI et al., 2025) | OpenAI | Medium (128K) | 1.0 | 1.0 |
| Grok-4-Fast-Reasoning (xAI, 2025) | xAI | 100K | 1.0 | 1.0 |
| Claude-Sonnet-4.5 (Anthropic, 2025) | Anthropic | 120K | 1.0 | 1.0 |
| SEED-OSS (ByteDance Seed Team, 2025) | ByteDance | Unlimited (128K) | 1.1 | 0.95 |
| Qwen3-32B (Yang et al., 2025b) | Alibaba | 38K | 0.6 | 0.95 |
| Qwen3-14B (Yang et al., 2025b) | Alibaba | 38K | 0.6 | 0.95 |
| QwQ-32B (Qwen Team, 2025) | Alibaba | 32K | 0.6 | 0.95 |
| Qwen3-30B (Yang et al., 2025b) | Alibaba | 38K | 0.6 | 0.95 |
| Qwen3-8B (Yang et al., 2025b) | Alibaba | 38K | 0.6 | 0.95 |
| Qwen3-4B (Yang et al., 2025b) | Alibaba | 38K | 0.6 | 0.95 |
| Gemini-2.5-Pro-exp-03-25 (Comanici et al., 2025) | Google | 64K | 1.0 | 0.95 |
| Gemini-2.5-Flash-preview-04-17 (Comanici et al., 2025) | Google | 64K | 1.0 | 0.95 |
| DeepSeek-R1-01-28 (DeepSeek-AI et al., 2025) | DeepSeek | 32K | 0.6 | 0.95 |
| DeepSeek-R1-Distill-Llama-70B (DeepSeek-AI et al., 2025) | DeepSeek | 32K | 0.6 | 0.95 |
| DeepSeek-R1-Distill-Qwen-32B (DeepSeek-AI et al., 2025) | DeepSeek | 32K | 0.6 | 0.95 |
| DeepSeek-R1-Distill-Qwen-14B (DeepSeek-AI et al., 2025) | DeepSeek | 32K | 0.6 | 0.95 |
| DeepSeek-R1-Distill-Llama-8B (DeepSeek-AI et al., 2025) | DeepSeek | 32K | 0.6 | 0.95 |

---

**Prompt Setup**

Below are the instructions we use to evaluate the models, and we also include this prompt template in our appendix.

Given a competition problem below, write a solution in C++ that solves all the subtasks. Make sure to wrap your code in '""<task>.cpp' and '"' Markdown delimiters.

[BEGIN PROBLEM]
<task statement>
[END PROBLEM]

Time limit: <time> seconds
Memory limit: <Memory> MB

Generate a solution in C++ that solves the task. Make sure to wrap your code in '""<task>.cpp' and '"' Markdown delimiters.

- - - - - - - - - - - - - - - - - - - - - - - - - - - - - - - - - - - - - - - - - - - - - - - - - - - -

Prompt configuration used in all experiments.

---

# C. Evaluation Metrics

- **Pass@k (Kulal et al., 2019; Chen et al., 2021):** We use the conventional Pass@k, which measures the fraction of problems for which at least one of the $k$ generated solutions is correct. We use $k = 8$.

- **Relative Score:** This metric is defined as the division of the model's score over the total possible score of a contest, providing a normalized measure of performance.

- **Average Percentile:** To benchmark LLM performance against human capabilities, we map the models' scores to a percentile rank based on the performance distribution of human contestants.

- **Olympiad Medal System:** It uses the authoritative cutoffs in the Olympiads to decide if a model's performance is qualified for a medal (gold, silver, or bronze).

- **Codeforces Elo:** Inspired by the widely used rating system in competitive programming, we treat each model as a "virtual contestant" and update its rating after every contest based on its relative standing against human participants.

# D. Full Results

We present the complete evaluation results of all models on LiveOIBench. Table 12 provides the overall leaderboard across all 72 contests, while Table 13 breaks down performance by contest tags.

*Table 12.* Main results of all models evaluated on 72 contests, with expanded Human Percentile, Pass Rate, and CF rating breakdowns.

| Model | Medals (%) | | | | Relative Score (%) | Human Percentile (%) | | | | | Pass Rate (%) | | | | CF Rating | | | | |
|---|---|---|---|---|---|---|---|---|---|---|---|---|---|---|---|---|---|---|---|
| | Gold | Silver | Bronze | Total | | All | D1 | D2 | D3 | D4 | All | Easy | Med. | Hard | All | D1 | D2 | D3 | D4 |
| **Proprietary LLMs** | | | | | | | | | | | | | | | | | | | |
| GPT-5 | 50.00 | 30.56 | 8.33 | 88.89 | 67.21 | 81.76 | 73.68 | 79.30 | 84.87 | 97.20 | 63.03 | 92.25 | 60.42 | 34.19 | 2414 | 2426 | 2322 | 2412 | 2583 |
| Grok-4-Fast-Reasoning | 45.83 | 20.83 | 16.67 | 83.33 | 56.99 | 74.23 | 63.79 | 67.62 | 79.14 | 97.46 | 50.95 | 70.42 | 39.58 | 15.38 | 2221 | 2053 | 2158 | 2224 | 2598 |
| Gemini-2.5-Pro | 31.94 | 22.22 | 23.61 | 77.78 | 51.33 | 71.80 | 55.84 | 65.23 | 79.02 | 95.90 | 44.46 | 82.39 | 32.64 | 16.24 | 2192 | 1963 | 2028 | 2308 | 2551 |
| GPT-O3-Mini-High | 26.39 | 23.61 | 22.22 | 72.22 | 47.69 | 64.28 | 53.07 | 50.76 | 75.79 | 94.67 | 44.19 | 84.51 | 34.03 | 11.11 | 2088 | 1807 | 1894 | 2284 | 2449 |
| Gemini-2.5-Flash | 15.28 | 23.61 | 23.61 | 62.50 | 41.29 | 56.81 | 43.48 | 43.79 | 69.89 | 93.97 | 36.06 | 75.35 | 22.92 | 6.84 | 1945 | 1700 | 1700 | 2091 | 2505 |
| Claude-Sonnet-4.5 | 11.11 | 20.83 | 34.72 | 66.67 | 38.30 | 53.08 | 36.69 | 49.80 | 65.79 | 85.63 | 27.05 | 61.97 | 9.72 | 5.13 | 1848 | 1766 | 1595 | 2057 | 2060 |
| GPT-4.1 | 4.17 | 13.89 | 22.22 | 40.28 | 24.78 | 35.99 | 21.47 | 24.90 | 47.56 | 79.73 | 18.32 | 48.59 | 4.86 | 3.42 | 1482 | 1339 | 1134 | 1724 | 1994 |
| **Open-weight Thinking LLMs** | | | | | | | | | | | | | | | | | | | |
| GPT-OSS-120B-High | 50.00 | 26.39 | 11.11 | 87.50 | 62.78 | 72.88 | 69.46 | 63.36 | 80.92 | 96.82 | 60.14 | 92.25 | 54.86 | 30.77 | 2205 | 1950 | 2122 | 2264 | 2520 |
| GPT-OSS-20B-High | 22.22 | 29.17 | 23.61 | 75.00 | 49.55 | 57.72 | 47.18 | 44.47 | 72.55 | 94.07 | 52.81 | 86.62 | 42.36 | 17.95 | 2020 | 1763 | 1797 | 2167 | 2504 |
| GPT-OSS-120B | 29.17 | 23.61 | 20.83 | 73.61 | 49.23 | 59.90 | 53.89 | 43.75 | 72.87 | 91.14 | 47.78 | 87.32 | 33.33 | 19.66 | 2032 | 1638 | 1894 | 2193 | 2493 |
| GPT-OSS-20B | 19.44 | 23.61 | 25.00 | 68.06 | 42.36 | 53.94 | 48.43 | 37.94 | 66.58 | 88.26 | 42.80 | 81.69 | 29.86 | 12.82 | 1901 | 1501 | 1660 | 2165 | 2383 |
| Qwen3-32B | 9.72 | 15.28 | 29.17 | 54.17 | 32.86 | 42.00 | 31.36 | 31.17 | 56.64 | 81.73 | 27.70 | 67.61 | 6.25 | 6.84 | 1665 | 1342 | 1455 | 1959 | 2022 |
| DeepSeek-R1 | 6.94 | 19.44 | 26.39 | 52.78 | 33.43 | 42.29 | 30.22 | 27.05 | 55.89 | 80.92 | 28.87 | 69.01 | 8.33 | 6.84 | 1617 | 1443 | 1278 | 1906 | 2015 |
| Qwen3-14B | 5.56 | 15.28 | 25.00 | 45.83 | 27.24 | 34.59 | 24.51 | 25.66 | 47.57 | 74.78 | 22.73 | 58.45 | 5.90 | 7.69 | 1402 | 976 | 1241 | 1652 | 1938 |
| QWQ-32B | 5.56 | 13.89 | 26.39 | 45.83 | 26.56 | 33.84 | 19.15 | 26.44 | 49.93 | 71.94 | 23.95 | 57.75 | 18.31 | 11.97 | 1491 | 1281 | 1113 | 1877 | 1956 |
| Qwen3-30B | 5.56 | 20.83 | 18.06 | 44.44 | 27.68 | 36.69 | 24.94 | 27.25 | 47.33 | 72.15 | 23.18 | 63.38 | 14.93 | 7.26 | 1549 | 1201 | 1323 | 1862 | 1995 |
| Qwen3-8B | 1.39 | 12.50 | 26.39 | 40.28 | 24.25 | 31.03 | 25.37 | 23.23 | 40.68 | 69.90 | 19.05 | 53.52 | 15.56 | 7.26 | 1426 | 1206 | 1312 | 1534 | 1789 |
| DeepSeek-R1-Distill-Llama-70B | 1.39 | 8.33 | 23.61 | 33.33 | 20.50 | 32.30 | 22.28 | 23.63 | 39.29 | 76.89 | 16.88 | 43.66 | 15.85 | 7.69 | 1283 | 1042 | 1103 | 1472 | 1665 |
| DeepSeek-R1-Distill-Qwen-32B | 1.39 | 8.33 | 20.83 | 30.56 | 19.14 | 27.03 | 11.43 | 15.59 | 37.55 | 44.75 | 14.86 | 40.14 | 1.39 | 1.71 | 1284 | 964 | 1074 | 1631 | 1549 |
| Qwen3-4B | 1.39 | 8.33 | 16.67 | 26.39 | 16.81 | 24.28 | 18.35 | 16.17 | 29.91 | 54.21 | 13.61 | 31.08 | 12.60 | 5.13 | 1153 | 970 | 897 | 1332 | 1622 |
| DeepSeek-R1-Distill-Qwen-14B | 1.39 | 2.78 | 9.72 | 13.89 | 13.41 | 22.77 | 15.33 | 15.23 | 25.64 | 47.24 | 10.56 | 25.35 | 9.15 | 4.05 | 1089 | 897 | 991 | 1166 | 1457 |
| DeepSeek-R1-Distill-Llama-8B | 0.00 | 0.00 | 2.78 | 2.78 | 3.10 | 11.86 | 10.82 | 10.37 | 9.51 | 33.46 | 2.46 | 10.56 | 1.56 | 0.00 | 724 | 724 | 628 | 705 | 1103 |
| **Open-weight Non-Thinking LLMs** | | | | | | | | | | | | | | | | | | | |
| DeepSeek-V3 | 4.17 | 8.33 | 22.22 | 34.72 | 21.70 | 31.76 | 16.42 | 20.86 | 41.88 | 73.73 | 17.10 | 43.66 | 2.78 | 2.56 | 1283 | 1239 | 1187 | 1598 | 1827 |
| Qwen3-32B-Non-Thinking | 1.39 | 4.17 | 11.11 | 16.67 | 12.92 | 24.64 | 14.14 | 17.35 | 24.10 | 52.37 | 8.78 | 21.13 | 0.69 | 4.27 | 1040 | 957 | 844 | 1227 | 1251 |
| Qwen2.5-Coder-32B-Instruct | 1.39 | 2.78 | 9.72 | 13.89 | 11.25 | 19.90 | 15.31 | 17.26 | 22.73 | 48.38 | 6.15 | 22.54 | 5.90 | 1.71 | 1023 | 983 | 701 | 1247 | 1384 |
| Qwen2.5-Coder-14B-Instruct | 1.39 | 2.78 | 6.94 | 11.11 | 9.66 | 19.56 | 13.48 | 14.56 | 23.55 | 46.20 | 5.53 | 20.42 | 4.51 | 1.71 | 966 | 935 | 849 | 969 | 1360 |
| Mistral-Large-Instruct-2411 | 1.39 | 1.39 | 8.33 | 11.11 | 9.99 | 18.70 | 14.62 | 15.13 | 22.66 | 43.03 | 5.90 | 21.13 | 4.65 | 1.71 | 1023 | 939 | 875 | 1122 | 1376 |
| Mistral-Small-3.1-24B-2503 | 1.39 | 0.00 | 9.72 | 11.11 | 7.75 | 19.08 | 10.67 | 10.92 | 20.04 | 42.86 | 4.75 | 17.61 | 3.41 | 1.71 | 909 | 805 | 822 | 879 | 1334 |
| Llama-4-Scout | 1.39 | 1.39 | 5.56 | 8.33 | 9.88 | 19.60 | 10.67 | 11.97 | 24.11 | 47.09 | 6.32 | 21.83 | 4.34 | 1.71 | 1008 | 825 | 892 | 1107 | 1316 |
| Qwen2.5-72B | 1.39 | 2.78 | 5.56 | 9.72 | 9.90 | 19.24 | 10.84 | 12.99 | 22.96 | 45.43 | 5.55 | 14.08 | 0.00 | 1.71 | 1000 | 875 | 862 | 1022 | 1508 |
| Llama-3.3-70B-Instruct | 0.00 | 1.39 | 8.33 | 9.72 | 10.00 | 21.37 | 15.04 | 16.67 | 26.37 | 50.32 | 5.65 | 20.42 | 4.79 | 1.71 | 1056 | 899 | 1069 | 1020 | 1458 |
| Qwen3-30B-Non-Thinking | 1.39 | 0.00 | 6.94 | 8.33 | 10.48 | 17.28 | 10.60 | 12.11 | 22.83 | 43.92 | 6.99 | 23.94 | 5.49 | 1.71 | 989 | 962 | 791 | 1052 | 1425 |
| Qwen3-14B-Non-Thinking | 0.00 | 1.39 | 5.56 | 6.94 | 6.65 | 15.30 | 6.65 | 8.76 | 10.91 | 44.36 | 4.47 | 10.56 | 0.00 | 2.56 | 894 | 818 | 753 | 932 | 1303 |
| Qwen3-8B-Non-Thinking | 0.00 | 1.39 | 2.78 | 4.17 | 7.53 | 16.82 | 9.78 | 9.01 | 14.16 | 39.65 | 4.04 | 10.56 | 0.00 | 1.71 | 843 | 745 | 701 | 842 | 1357 |
| CODESTRAL-22B-V0.1 | 0.00 | 1.39 | 2.78 | 4.17 | 6.84 | 15.94 | 9.81 | 7.46 | 6.86 | 42.39 | 4.34 | 8.45 | 1.39 | 2.56 | 912 | 948 | 784 | 895 | 1275 |
| Llama-3.1-8B-Instruct | 0.00 | 1.39 | 1.39 | 2.78 | 4.19 | 13.49 | 7.90 | 6.15 | 7.60 | 30.13 | 2.45 | 5.63 | 0.00 | 0.85 | 761 | 714 | 644 | 808 | 1073 |

*Table 13.* Pass rate of all tags for each model, from easiest to hardest based on difficulty labels. Abbreviations: IM (implementation), MA (mathematics), AH (ad-hoc), PS (prefix sum), SO (sorting), GR (greedy), GTR (graph traversal), BS (binary search), NT (number theory), GT (graph theory), DS (data structures), CB (combinatorics), DP (dynamic programming), TR (tree), ST (segment tree).

| Model | IM | MA | AH | PS | SO | GR | GTR | BS | NT | GT | DS | CB | DP | TR | ST |
|---|---|---|---|---|---|---|---|---|---|---|---|---|---|---|---|
| **Proprietary LLMs** | | | | | | | | | | | | | | | |
| GPT-5 | 71.79 | 71.43 | 43.48 | 73.33 | 75.56 | 60.00 | 71.43 | 54.84 | 64.71 | 66.67 | 66.27 | 64.71 | 46.88 | 37.50 | 56.41 |
| Gemini-2.5-Pro | 66.67 | 71.43 | 30.43 | 53.33 | 57.78 | 37.14 | 42.86 | 38.71 | 35.29 | 44.44 | 38.55 | 58.82 | 23.44 | 20.83 | 30.77 |
| GPT-O3-Mini-High | 64.10 | 71.43 | 34.78 | 46.67 | 60.00 | 37.14 | 46.43 | 41.94 | 41.18 | 38.89 | 38.55 | 47.06 | 34.38 | 20.83 | 28.21 |
| Gemini-2.5-Flash | 64.10 | 71.43 | 30.43 | 46.67 | 48.89 | 28.57 | 25.00 | 32.26 | 29.41 | 29.63 | 30.12 | 47.06 | 20.31 | 12.50 | 15.38 |
| GPT-4.1 | 53.85 | 50.00 | 26.09 | 40.00 | 13.33 | 14.29 | 7.14 | 12.90 | 17.65 | 12.96 | 12.05 | 29.41 | 6.25 | 4.17 | 5.13 |
| **Open-weight Thinking LLMs** | | | | | | | | | | | | | | | |
| GPT-OSS-120B-High | 71.79 | 71.43 | 39.13 | 73.33 | 82.22 | 57.14 | 75.00 | 51.61 | 58.82 | 55.56 | 62.65 | 58.82 | 46.88 | 41.67 | 51.28 |
| GPT-OSS-120B-Medium | 64.10 | 64.29 | 34.78 | 53.33 | 60.00 | 40.00 | 53.57 | 38.71 | 41.18 | 44.44 | 44.58 | 58.82 | 35.94 | 25.00 | 35.90 |
| GPT-OSS-120B-Low | 61.54 | 71.43 | 30.43 | 46.67 | 37.78 | 31.43 | 35.71 | 29.03 | 23.53 | 27.78 | 27.71 | 47.06 | 17.19 | 16.67 | 15.38 |
| GPT-OSS-20B-High | 69.44 | 76.92 | 50.00 | 64.29 | 73.81 | 53.57 | 51.85 | 48.15 | 46.67 | 44.23 | 50.70 | 53.33 | 50.00 | 40.00 | 48.48 |
| GPT-OSS-20B-Medium | 63.16 | 71.43 | 40.91 | 57.14 | 51.11 | 36.36 | 35.71 | 36.67 | 47.06 | 30.19 | 36.59 | 66.67 | 29.69 | 22.73 | 26.32 |
| GPT-OSS-20B-Low | 56.41 | 64.29 | 30.43 | 40.00 | 33.33 | 17.14 | 25.00 | 23.33 | 23.53 | 27.78 | 24.69 | 35.29 | 17.46 | 12.50 | 13.16 |
| Seed-OSS | 61.54 | 64.29 | 36.36 | 53.33 | 48.89 | 31.43 | 32.14 | 38.71 | 35.29 | 27.78 | 34.94 | 52.94 | 26.56 | 12.50 | 28.21 |
| Qwen3-32B | 58.97 | 61.54 | 30.43 | 35.71 | 28.89 | 21.88 | 21.43 | 16.67 | 29.41 | 22.64 | 22.22 | 29.41 | 14.29 | 4.35 | 8.11 |
| DeepSeek-R1 | 61.54 | 64.29 | 30.43 | 33.33 | 28.89 | 17.14 | 17.86 | 22.58 | 29.41 | 22.22 | 20.48 | 29.41 | 15.62 | 4.17 | 7.69 |
| Qwen3-14B | 51.28 | 61.54 | 26.09 | 35.71 | 24.44 | 15.62 | 14.29 | 13.33 | 29.41 | 18.87 | 19.75 | 35.29 | 12.70 | 4.35 | 5.41 |
| QWQ-32B | 53.85 | 61.54 | 26.09 | 28.57 | 26.67 | 15.62 | 10.71 | 20.00 | 23.53 | 15.09 | 13.58 | 29.41 | 14.29 | 4.35 | 5.41 |
| Qwen3-30B | 43.59 | 61.54 | 26.09 | 28.57 | 31.11 | 18.75 | 28.57 | 13.33 | 29.41 | 24.53 | 23.46 | 41.18 | 15.87 | 4.35 | 5.41 |
| Qwen3-8B | 33.33 | 57.14 | 17.39 | 26.67 | 8.89 | 5.71 | 0.00 | 9.68 | 29.41 | 9.26 | 13.25 | 35.29 | 10.94 | 4.17 | 2.56 |
| DeepSeek-R1-Distill-Llama-70B | 41.03 | 50.00 | 17.39 | 20.00 | 20.00 | 17.14 | 10.71 | 16.13 | 17.65 | 14.81 | 13.25 | 11.76 | 9.38 | 4.17 | 5.13 |
| DeepSeek-R1-Distill-Qwen-32B | 38.46 | 46.15 | 21.74 | 14.29 | 15.56 | 12.50 | 7.14 | 10.00 | 11.76 | 5.66 | 8.64 | 11.76 | 3.17 | 0.00 | 2.70 |
| Qwen3-4B | 46.15 | 50.00 | 17.39 | 13.33 | 15.56 | 8.57 | 10.71 | 9.68 | 11.76 | 11.11 | 9.64 | 17.65 | 4.69 | 4.17 | 2.56 |
| DeepSeek-R1-Distill-Qwen-14B | 33.33 | 46.15 | 8.70 | 0.00 | 13.33 | 6.25 | 7.14 | 10.00 | 5.88 | 9.43 | 6.17 | 5.88 | 1.59 | 4.35 | 0.00 |
| DeepSeek-R1-Distill-Llama-8B | 12.82 | 23.08 | 0.00 | 0.00 | 0.00 | 0.00 | 0.00 | 0.00 | 0.00 | 0.00 | 0.00 | 0.00 | 0.00 | 0.00 | 0.00 |
| **Open-weight Non-Thinking LLMs** | | | | | | | | | | | | | | | |
| DeepSeek-V3 | 51.28 | 46.15 | 21.74 | 28.57 | 20.00 | 12.50 | 14.29 | 13.33 | 17.65 | 15.09 | 14.81 | 11.76 | 7.94 | 8.70 | 8.11 |
| Qwen3-32B-Non-Thinking | 25.64 | 42.86 | 13.04 | 0.00 | 6.67 | 5.71 | 3.57 | 9.68 | 11.76 | 7.41 | 2.41 | 11.76 | 4.69 | 0.00 | 2.56 |
| Qwen2.5-Coder-32B-Instruct | 25.64 | 46.15 | 8.70 | 0.00 | 6.67 | 6.25 | 3.57 | 6.67 | 11.76 | 3.77 | 4.94 | 5.88 | 3.17 | 0.00 | 0.00 |
| Qwen2.5-Coder-14B-Instruct | 20.51 | 46.15 | 9.09 | 0.00 | 6.82 | 3.12 | 3.57 | 3.33 | 5.88 | 5.77 | 1.23 | 11.76 | 1.61 | 0.00 | 0.00 |
| Mistral-Large-Instruct-2411 | 28.21 | 42.86 | 13.04 | 0.00 | 4.44 | 0.00 | 3.57 | 9.68 | 5.88 | 3.70 | 1.20 | 11.76 | 3.12 | 0.00 | 0.00 |
| Mistral-Small-3.1-24B-2503 | 23.08 | 46.15 | 8.70 | 0.00 | 4.44 | 0.00 | 3.57 | 3.33 | 5.88 | 3.77 | 2.47 | 5.88 | 1.59 | 0.00 | 0.00 |
| Qwen2.5-72B | 23.08 | 38.46 | 9.09 | 0.00 | 6.82 | 3.12 | 3.57 | 6.67 | 5.88 | 1.92 | 2.47 | 0.00 | 1.61 | 0.00 | 0.00 |
| Llama-3.3-70B-Instruct | 23.08 | 38.46 | 8.70 | 0.00 | 6.67 | 3.12 | 3.57 | 3.33 | 5.88 | 5.66 | 1.23 | 5.88 | 1.59 | 0.00 | 0.00 |
| Qwen3-30B-Non-Thinking | 23.68 | 30.77 | 8.70 | 6.67 | 8.89 | 2.86 | 7.14 | 6.45 | 5.88 | 9.43 | 2.44 | 17.65 | 0.00 | 0.00 | 0.00 |
| Qwen3-4B-Non-Thinking | 28.21 | 42.86 | 8.70 | 0.00 | 4.44 | 0.00 | 3.57 | 6.45 | 5.88 | 5.56 | 1.20 | 5.88 | 1.56 | 0.00 | 0.00 |
| Qwen3-8B-Non-Thinking | 20.51 | 30.77 | 4.35 | 0.00 | 8.89 | 2.86 | 0.00 | 3.23 | 5.88 | 5.56 | 1.20 | 0.00 | 0.00 | 0.00 | 0.00 |
| Codestral-22B-V0.1 | 20.51 | 38.46 | 4.35 | 0.00 | 2.22 | 0.00 | 3.57 | 0.00 | 0.00 | 7.55 | 0.00 | 0.00 | 1.59 | 0.00 | 0.00 |
| Llama-3.1-8B-Instruct | 15.38 | 38.46 | 4.35 | 0.00 | 2.22 | 0.00 | 3.57 | 3.33 | 5.88 | 1.89 | 1.23 | 0.00 | 0.00 | 0.00 | 0.00 |
| Qwen3-14B-Non-Thinking | 18.75 | 18.18 | 9.52 | 0.00 | 13.95 | 5.88 | 7.69 | 6.45 | 6.25 | 5.88 | 3.75 | 0.00 | 1.61 | 0.00 | 0.00 |
| DeepSeek-Coder-V2-Lite-Instruct | 12.82 | 30.77 | 0.00 | 0.00 | 0.00 | 0.00 | 3.57 | 3.33 | 0.00 | 3.77 | 0.00 | 0.00 | 0.00 | 0.00 | 0.00 |
| Qwen2.5-Coder-7B-Instruct | 13.16 | 35.71 | 8.70 | 0.00 | 2.22 | 0.00 | 3.57 | 3.45 | 5.88 | 2.04 | 1.23 | 0.00 | 1.59 | 0.00 | 0.00 |

## E. Additional Analysis

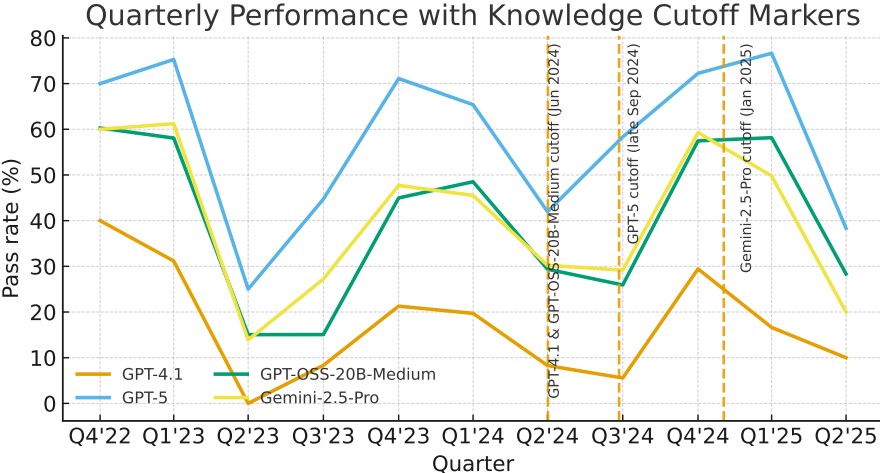

*Figure 5.* Mainstream model performance over quarters. This plot shows consistent performance trends among selected models and confirms no data contamination in mainstream LLMs.

### E.1. Model Performance across Years

Figure 5 shows quarterly pass rates of four mainstream LLMs from Q4'22 to Q2'25. The performance trends are broadly similar across models: all experience an early decline in 2023, recover through 2024, peak around late 2024 to early 2025, and then drop again in Q2'25. Importantly, there is no sharp bump or drop around the knowledge cutoff, suggesting that these models are not facing significant data contamination issues. Quantitatively, GPT-5 consistently leads: in its stronger quarters (Q1'23, Q4'23, Q1'25), it consistently outperforms Gemini-2.5-Pro and GPT-OSS-20B-Medium by about 15–25 percentage points, which is in line with Table 12.

### E.2. Inference-Time Scaling

Inference-time scaling has been shown effective for improving model performance in math (Snell et al., 2025; Brown et al., 2024) and coding (Li et al., 2025a; Ehrlich et al., 2025) domains. We investigate two dimensions: *parallel scaling* involves sampling multiple diverse solution candidates (Chen et al., 2021; Jain et al., 2025), while *sequential scaling* generates long chains-of-thought with complex reasoning strategies such as self-reflection and backtracking (DeepSeek-AI et al., 2025).

**Parallel Scaling: GPT-5 demonstrates a superior coding capacity boundary.** Figure 6a reveals significant differences in coding capacity boundaries (Yue et al., 2025) across models as measured by Pass@k. GPT-5 could pass around 64% of the problems given 8 attempts per problem. The steepest improvements occur between Pass@1 and Pass@4, indicating that the marginal benefit of additional attempts diminishes rapidly as models approach their capacity limits (Kulal et al., 2019). The persistent performance gaps between proprietary and open-source models across all sampling levels suggest fundamental differences in maximum coding capability rather than artifacts of insufficient attempts (Li et al., 2022b; Hendrycks et al., 2021).

**Sequential Scaling: Reasoning models benefit from additional reasoning token budget.** Figure 6b shows pass rates improving as token budget increases across all three models. GPT-OSS-120B achieves the highest performance with the fewest tokens generated. A key insight emerges: smaller models can approach larger model performance with sufficient reasoning budget, suggesting a practical trade-off for resource-constrained practitioners who may prefer specialized smaller models over large ones.

Both scaling approaches provide complementary benefits but face efficiency limitations. Sequential scaling shows promise for complex algorithmic problems but requires substantial computational resources, while parallel scaling reveals each model's performance ceiling as improvements plateau with additional samples (Chen et al., 2021; Austin et al., 2021). Future work could focus on developing hybrid approaches that combine both scaling paradigms while reducing computational overhead.

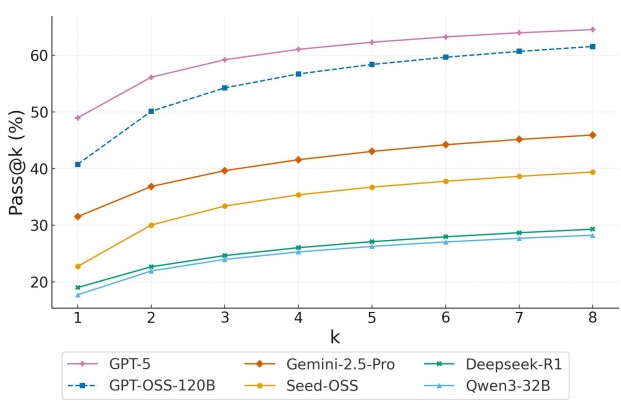

*(a)* **Parallel Scaling.** Pass@k vs. number of samples per problem.

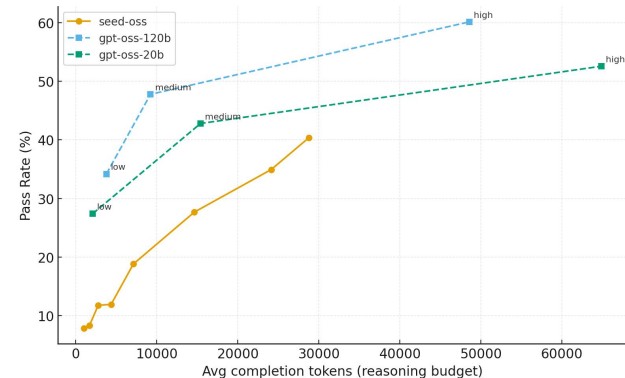

*(b)* **Sequential Scaling.** Pass rate vs. reasoning budget.

*Figure 6.* **Parallel vs. Sequential Scaling.** *Left*: Increasing the number of sampled solutions (k) improves Pass@k, with GPT-5 achieving the highest sample efficiency and performance ceiling. *Right*: Increasing reasoning budget improves performance, with models exhibiting different token efficiencies.

*Table 14.* Performance comparison of GPT-OSS-120B across Python, Java, and C++ on 22 USACO competitive programming contests. Metrics shown include relative score, pass rate, and medal distributions (gold, silver, bronze, and total medal percentage). Results indicate that C++ achieves the highest overall performance, followed by Java, with Python exhibiting significantly lower performance, likely due to differences in execution speed, memory management, and efficiency of standard libraries.

| Model | Relative Score (%) | Pass Rate (%) | Gold (%) | Silver (%) | Bronze (%) | Total Medals (%) |
|---|---|---|---|---|---|---|
| GPT-OSS-120B_Python | 44.35 | 46.97 | 13.64 | 13.64 | 27.27 | 54.55 |
| GPT-OSS-120B_Java | 50.78 | 56.82 | 9.09 | 36.36 | 27.27 | 72.73 |
| GPT-OSS-120B_C++ | 53.26 | 59.09 | 27.27 | 31.82 | 13.64 | 72.73 |

### E.3. Performance Variations Across Programming Languages

To provide deeper insights into language-specific model capabilities, we present a comparative analysis of GPT-OSS-120B's performance across Python, Java, and C++, evaluated on a subset of 132 problems from the USACO contest series.

As shown in Table 14, the model performs best when generating solutions in C++, highlighting the inherent advantage of C++ in competitive programming due to its superior execution speed, precise memory control, and optimized functions. Java delivers intermediate performance; although robust and widely supported, it incurs overhead from JVM execution and automated memory management, placing it slightly behind C++. Meanwhile, although Python is the most commonly used and extensively studied language for evaluating LLM coding capability, it shows a considerable performance gap due to its interpreted nature and dynamic typing, sacrificing critical speed and efficiency. Thus, contest results clearly show C++ performing best, followed by Java, with Python significantly behind.

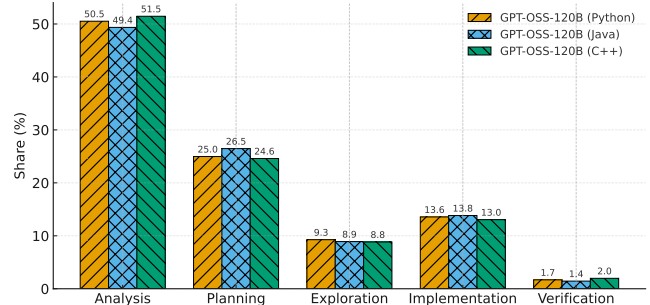

*Figure 7.* Histogram showing that reasoning behaviors across Python, Java, and C++ are similar, suggesting that the performance gain from using C++ mainly comes from the language efficiency.

To better understand the reasons behind these performance differences, we further analyzed GPT-OSS-120B's reasoning behaviors across these programming languages. As illustrated in the Figure 7, the model consistently allocates reasoning tokens similarly across Python, Java, and C++, indicating that its reasoning strategy remains stable irrespective of language choice. This stability in reasoning behavior suggests that the performance advantage observed for C++ primarily stems from computational efficiency rather than differences in the underlying reasoning approach.

## E.4. Example of Semantic Rephrasing

To generate semantically equivalent task variants for our contamination check, we rewrite task narratives, object names, and illustrative strings while preserving the underlying algorithmic structure. The example below shows how the surface form is changed without changing the core operation: extracting repeated target strings as subsequences from a noisy input.

---

**Semantic Rephrasing Example**

**Original snippet.** Pareidolia is the phenomenon where your eyes tend to see familiar patterns in images where none really exist—for example, seeing a face in a cloud. As you might imagine, with John's constant proximity to cows, he often sees cow-related patterns in everyday objects. For example, if he looks at the string `"bqessiyexbesszieb"`, John's eyes ignore some of the letters and all he sees is `"bessiebessie"`.

**Rephrased snippet.** In a deep-space signal archive, analysts search noisy transmissions for repeated occurrences of the beacon code `"orion7"`. A transmission may contain many extra symbols, and analysts are allowed to ignore some of them. For example, from the string `"xoriqon7morionk7z"`, they can ignore certain characters and perceive only `"orion7orion7"`.

---

## E.5. Impact of Template Similarity on Model Performance

To investigate whether models exhibit similar performance on problems sharing similar solution templates, we compute the official solution similarity for each problem pair and analyze its relationship with the models' performance differences on these problems. Specifically, we assessed GPT-5 and Grok-4-Fast-Reasoning by plotting the absolute difference in their relative scores against each problem pair's solution similarity (Figure 8). If template-based contamination significantly influenced model performance, we would expect to observe consistently small performance differences between problem pairs with similar official solutions. However, our analysis reveals only a very weak and negligible correlation (GPT-5: Pearson's r = -0.084; Grok-4: Pearson's r = -0.072). This indicates that even problem pairs with similar official solution structures often result in substantially different model performance. Additionally, we found that most problem pairs exhibit low similarity, emphasizing the wide variety of problem-solving approaches necessary for success on our benchmark.

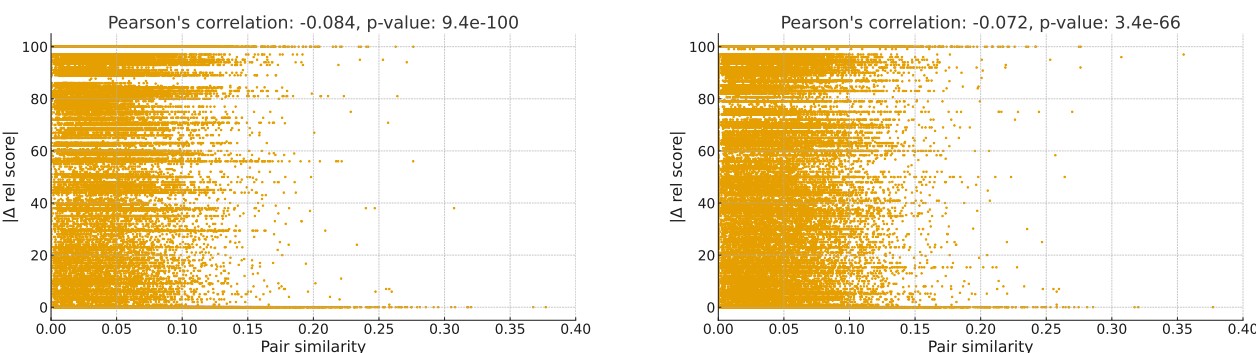

*Figure 8.* Scatter plots of solution similarity vs. model performance differences for GPT-5 (left) and Grok-4 (right). Weak correlations (GPT-5: $r = -0.08$; Grok-4: $r = -0.07$) indicate minimal template-based contamination.

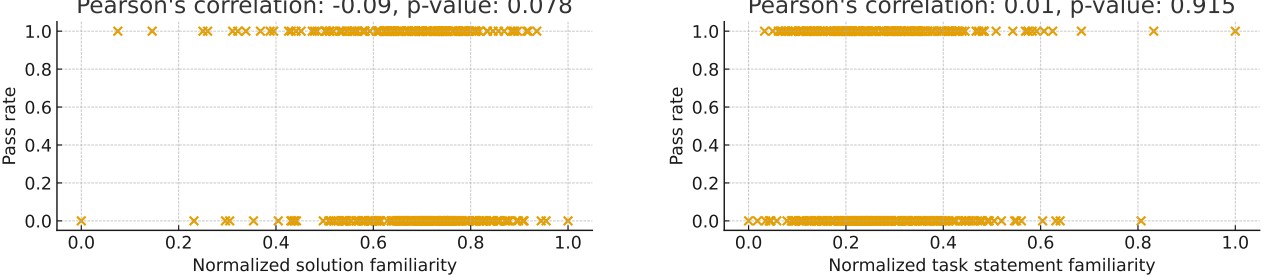

*Figure 9.* No significant positive correlation is observed between GPT-OSS-120B's familiarity with task statements and solutions (normalized via min-max scaling) and its performance, indicating that higher familiarity does not necessarily translate to better outcomes.

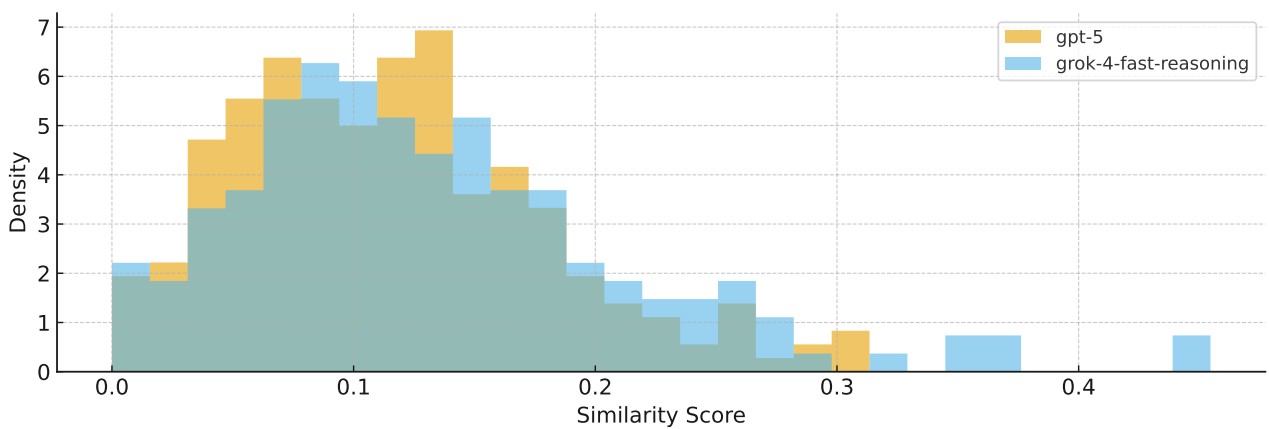

*Figure 10.* Histogram showing similarity scores between accepted model solutions (GPT-5 and Grok-4-Fast-Reasoning) and their corresponding official solutions. All similarity scores remain below 50%, indicating that accepted model solutions are substantially distinct from official solutions. Density represents the probability per unit similarity, so that the total area of the histogram equals 1.

*Table 15.* Percentage improvements across algorithmic task categories from scaling model parameters and reasoning-token budgets.

| Model Δ (%) | IM | MA | AH | PS | SO | GR | GTR | BS | NT | GT | DS | CB | DP | TR | ST | Avg |
|---|---|---|---|---|---|---|---|---|---|---|---|---|---|---|---|---|
| GPT-20B-Low → GPT-120B-Low | +9.10 | +11.10 | +0.00 | +16.70 | +13.40 | +83.30 | +42.80 | +24.40 | +0.00 | +0.00 | +12.20 | +33.40 | -1.50 | +33.40 | +16.90 | **18.20** |
| GPT-20B-Low → GPT-20B-High | +23.06 | +19.64 | +64.28 | +60.73 | +121.44 | +212.64 | +107.40 | +106.46 | +98.36 | +59.21 | +105.40 | +51.09 | +186.43 | +220.00 | +268.33 | **113.47** |
| Qwen3-14B → GPT-20B-Medium | +23.17 | +16.07 | +56.77 | +59.94 | +109.08 | +132.79 | +149.86 | +174.99 | +60.00 | +60.00 | +85.19 | +88.91 | +133.46 | +322.07 | +286.33 | **118.06** |

## E.6. Additional Analysis of How Different Attributes Affect Model Performance

In Table 13 and Table 15, we present a detailed analysis of model performance across various algorithmic categories, examining three primary factors: model size, reasoning-token budget, and training strategies (RL and distillation).

**Model Size**: Scaling GPT-OSS from 20B to 120B parameters under low-reasoning effort yields an average improvement of +18%. This gain is concentrated mainly in Greedy (+83.30%), Graph Traversal (+42.80%), Combinatorics (+33.40%), and Tree (+33.40%). These patterns indicate that additional parameters enhance the model's structural representations and pattern-recognition capacity, enabling better handling of connectivity and global relational structure.

**Reasoning Tokens**: Increasing the reasoning-token budget from low to high, while maintaining the same GPT-OSS-20B model weights, produces a substantially larger average improvement of +113%. Gains are most pronounced in algorithmic tasks such as segment tree (+268.33%), tree traversal (+220.00%), greedy algorithms (+212.64%), dynamic programming (+186.43%), and sorting (+121.44%), all of which demand extended multi-step, sequential reasoning. Notably, greedy and tree traversal appear in the top-5 for both factors: larger models recognize their structural patterns more reliably, while deeper reasoning is essential to execute the multi-step, stateful chains these problems demand.

**RL vs. Distillation**: Since each model family differs in architecture, pretraining data, and training pipelines, it's difficult to cleanly isolate the effects of RL versus distillation on downstream performance. However, preliminary comparisons between Qwen3-14B (a distilled model) and GPT-OSS-20B (trained via SFT and RL) suggest RL-trained models, with SFT warmup and enough RL training, can substantially outperform distilled models, with an average improvement of +118%. Isolating the contribution of each training stage is an important future direction for understanding how these methods shape algorithmic skills.

Overall, our analyses show that reasoning token allocation, far more than model size, drives improvements on the most algorithmically complex tasks. Also, directly trained RL models perform better than distilled models.

## E.7. Prompt Sensitivity

We generate four alternative prompt variants using GPT-5 and run GPT-OSS-120B-Medium with four additional prompt variants to explore how different instructions might influence the model's performance. While these prompts do not represent all possible instruction styles, we observe minimal performance differences across them, suggesting that prompt variants

*Table 16.* Performance of GPT-OSS-120B-Medium under different prompt variants.

| Model | Relative Score (%) | Pass Rate (%) | Human Percentile |
|---|---|---|---|
| GPT-OSS-120B-P1 | 47.76 | 45.85 | 58.50 |
| GPT-OSS-120B-P2 | 48.07 | 47.10 | 59.76 |
| GPT-OSS-120B-P3 | 46.75 | 45.69 | 57.73 |
| GPT-OSS-120B-P4 | 48.04 | 45.68 | 59.49 |
| GPT-OSS-120B-Medium | 48.88 | 47.33 | 58.10 |
| **Mean ± Std** | **47.90 ± 0.69** | **46.33 ± 0.73** | **58.72 ± 0.79** |

have little influence on model performance. The default prompt and prompt variants can be found in Table B and Table E.7.

In Table 16, across all five prompt variants, GPT-OSS-120B achieves $47.9 \pm 0.7\%$ relative score and $46.3 \pm 0.7\%$ pass rate, with human percentile tightly clustered around $58.7 \pm 0.8$, indicating low sensitivity to prompt choices. A more systematic and large-scale study of prompt sensitivity is an interesting direction for future work.

---

**Prompt Variants**

**Prompt 1.** You are a helpful and knowledgeable AI assistant. Answer the user's questions clearly and concisely, using correct reasoning and accurate information.

**Prompt 2.** You are an expert IOI competitive programming assistant. For each problem, output only a single complete C++17 solution that compiles and solves the task. Do not include explanations, comments, or any text outside one `cpp` code block. Read input from standard input and write output to standard output.

**Prompt 3.** You are an expert IOI problem solver. First, think through the solution step by step and explain your reasoning in a few clear paragraphs or bullet points. After that, output a single complete C++17 solution inside one `cpp` code block that follows your reasoning, reads from standard input, and writes to standard output.

**Prompt 4.** You are a meticulous and safe IOI problem-solving assistant. Always prioritize correctness, handling of corner cases, and numerical stability over constant-factor performance. Carefully verify your reasoning before producing an answer. When writing code, output a single complete C++17 solution in a `cpp` code block, making sure it compiles, handles invalid or extreme inputs robustly when appropriate, and does not perform unsafe or undefined operations.

---

### E.8. Analysis of Template-Based Contamination and Model Performance

We selected CodeContest (Li et al., 2022a; Wang et al., 2025) as our seed dataset to represent template solutions commonly found in training data, as it is among the most prominent competitive coding datasets with over 10,000 problems. For each problem, we sampled three C++ solutions, filtering out solutions shorter than 20 lines or longer than 1,000 lines. Using Dolos, we calculated the similarity between official solutions and these template solutions, recording the highest similarity score for each official solution. A higher similarity score indicates a greater likelihood of template-based contamination.

To examine whether template similarity influences model performance, we conducted correlation analyses using GPT-5 and Grok-4-Fast-Reasoning. In Figure 11, our analysis revealed weak correlations between template similarity and relative performance scores: GPT-5 exhibited a Pearson correlation coefficient of 0.136, and Grok-4 demonstrated a Pearson correlation coefficient of 0.193. These correlations, though slightly stronger than those observed in our previous analysis, remain limited.

This finding indicates that template similarity contributes minimally to model performance. Rather, the dominant factor influencing performance is the model's intrinsic problem-solving and reasoning capabilities. Given the nature of competitive coding problems, models rely on learned algorithmic strategies and analytical techniques to solve problems effectively. Therefore, we contend that the minimal template-based contamination observed does not undermine the validity of our evaluations. Models must effectively utilize learned skills from their training, applying them innovatively to solve new and complex challenges.

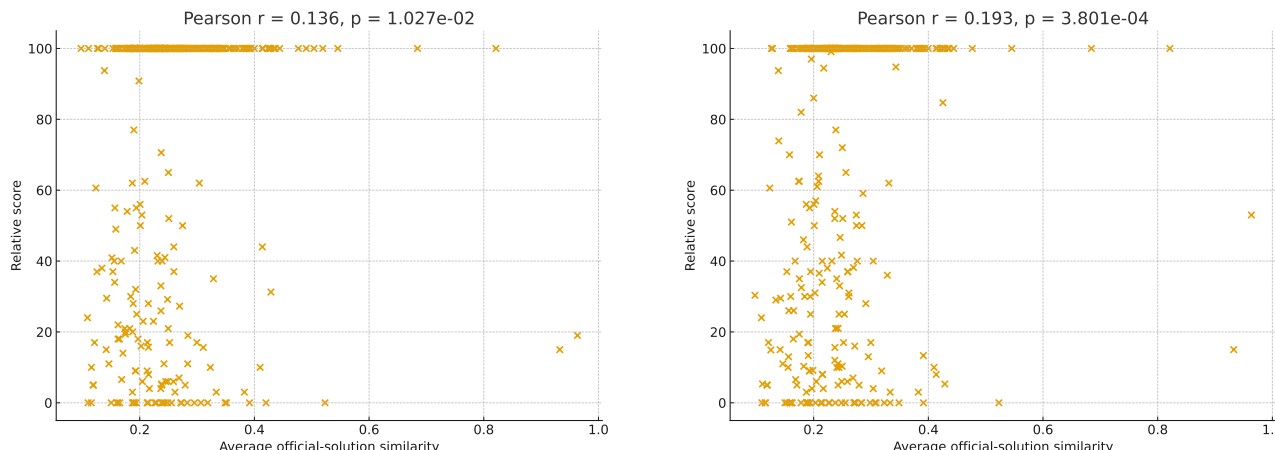

*Figure 11.* Scatter plots illustrating the relationship between official-solution similarity and relative scores for GPT-5 (left) and Grok-4-Fast-Reasoning (right). Both models exhibit weak correlations (Pearson r = 0.136, p = 0.010 for GPT-5; Pearson r = 0.193, p < 0.001 for Grok-4), suggesting minimal impact of template similarity on overall model performance.

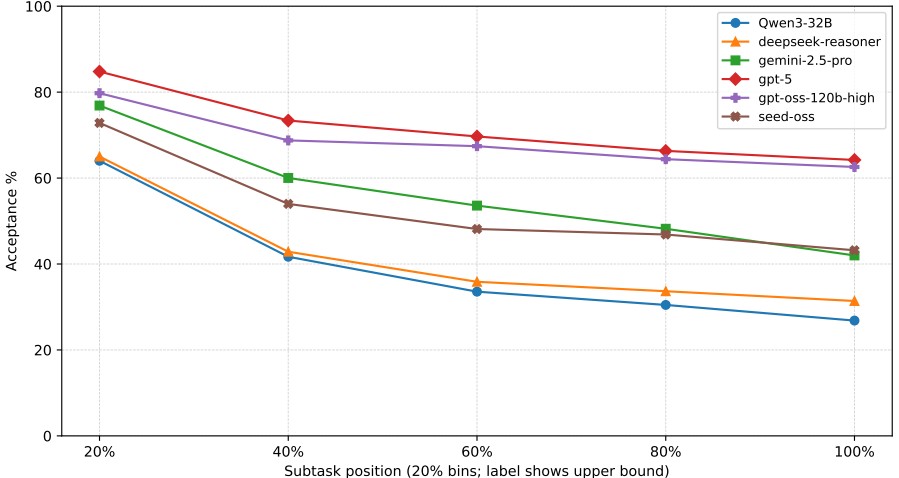

*Figure 12.* Mainstream model performance over subtask positions. As expected, later subtasks pose greater challenges for LLMs.

## E.9. Additional Discussion on Future Directions

Specifically, guiding models to focus on structured planning and detailed analysis, while limiting unnecessary exploration, can greatly improve solution accuracy and efficiency. Additionally, developing mechanisms that help models internalize verification during planning and analysis phases could reduce their dependence on explicit, post-hoc verification steps. Finally, creating automated techniques to dynamically adjust reasoning budgets and behaviors according to problem-specific factors like difficulty and algorithms may further boost the effectiveness of reasoning models in solving complex competitive programming tasks.

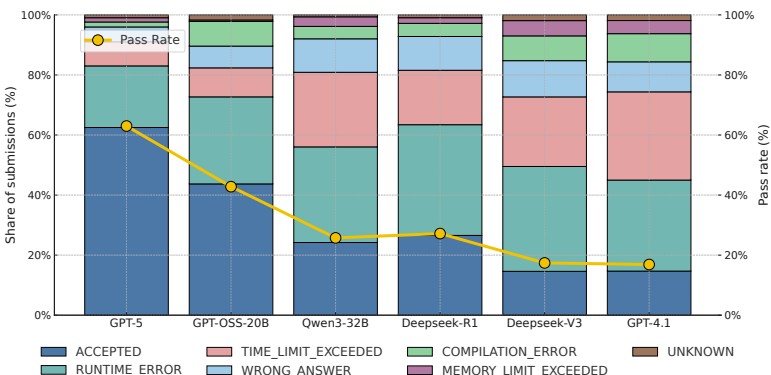

*Figure 13.* Submission status distribution for six selected models. The models are sorted based on performance from left to right. Solutions by stronger reasoning models show substantial reductions in failure types of time limit, memory limit, and compilation errors.

### E.10. Reasoning Behaviors Analysis

As described in Section 5, we partition each reasoning trace into segments of approximately 5k tokens, estimated by dividing the total token length by four. We categorize models' reasoning traces into eight behaviors, which we group into five broader categories: **Analysis** (Algorithm/Proof Analysis, Complexity Analysis), **Planning** (Problem Restatement, Subgoal Setting), **Exploration** (Backtracking, Dead-end Recognition), **Implementation** (Pseudo Implementation), and **Verification** (Test Case Verification).

The following prompts were used to elicit and analyze these reasoning behaviors of each segment:

- `PR_PROMPT` → Problem Restatement (Planning). See Prompt 1.

- `CMP_PROMPT` → Complexity Analysis (Analysis). See Prompt 2.

- `VT_PROMPT` → Test Case Verification (Verification). See Prompt 3.

- `SUB_PROMPT` → Subgoal Setting (Planning). See Prompt 4.

- `DED_PROMPT` → Dead-end Recognition (Exploration). See Prompt 5.

- `BKT_PROMPT` → Backtracking (Exploration). See Prompt 6.

- `AP_PROMPT` → Algorithm/Proof Analysis (Analysis). See Prompt 7.

- `PSD_PROMPT` → Pseudo Implementation (Implementation). See Prompt 8.

```
PR_PROMPT = """
You are an auditor. Count occurrences of the behavior PR (Problem Restatement) in a
    competitive-programming reasoning trace.

DEFINITION (apply strictly)
PR = Expressing the task in the solver's own words to clarify WHAT must be computed/
    decided/constructed (not HOW).
Include: restating the goal/output/validity conditions; clarifying what constitutes a
    correct answer.

COUNT
- Count 1 per PR-labeled step.

OUTPUT (strict JSON ONLY -- no extra text):
{
  "PR": <integer count>,
  "events": [
    {"snippet": "<short quote>", "reason": "<why it matches PR>"}
  ]
}

<TRACE>
{TRACE}
</TRACE>

Analyze the trace and count the occurrences of PR.
"""
```

Prompt 1: PR_PROMPT (Problem Restatement)

```
CMP_PROMPT = """
You are an auditor. Count occurrences of the behavior CMP (Complexity Analysis) in a
    competitive-programming reasoning trace.
DEFINITION
CMP = Analyzing asymptotic time/space complexity and feasibility versus constraints.

COUNT
- Count 1 per CMP-labeled step.

OUTPUT (strict JSON ONLY):
{
  "CMP": <integer count>,
  "events": [
    {"snippet": "<short quote>", "reason": "<why it matches CMP>"}
  ]
}

<TRACE>
{TRACE}
</TRACE>

Analyze the trace and count the occurrences of CMP.
"""
```

Prompt 2: CMP_PROMPT (Complexity Analysis)

```
VT_PROMPT = """
You are an auditor. Count occurrences of the behavior V-T (Test Cases Verification) in
     a competitive-programming reasoning trace.

DEFINITION
V-T = Checking the method on specific inputs and comparing with expected/reference
    outcomes.
Include: "On sample 2, expected=5, we get 5"; "Fails on [3,3,2] with output 7".

COUNT
- Count 1 per V-T-labeled step (multiple tests in one step = 1).

OUTPUT (strict JSON ONLY):
{
  "V-T": <integer count>,
  "events": [
    {"snippet": "<short quote>", "reason": "<why it matches V-T>"}
  ]
}

<TRACE>
{TRACE}
</TRACE>

Analyze the trace and count the occurrences of V-T.
"""
```

Prompt 3: VT_PROMPT (Test Case Verification)

```
SUB_PROMPT = """
You are an auditor. Count occurrences of the behavior SUB (Subgoal Setting) in a
```

```
        competitive-programming reasoning trace.

DEFINITION
SUB = Breaking the solution into intermediate objectives or a checklist before
    implementation.
Include: ordered lists like "parse -> preprocess -> compute -> output"; milestones
    like "build graph; find components; count sizes".

COUNT
- Count 1 per SUB-labeled step.

OUTPUT (strict JSON ONLY):
{
  "SUB": <integer count>,
  "events": [
    {"snippet": "<short quote>", "reason": "<why it matches SUB>"}
  ]
}

<TRACE>
{TRACE}
</TRACE>

Analyze the trace and count the occurrences of SUB.
"""
```

Prompt 4: SUB_PROMPT (Subgoal Setting)

```
DED_PROMPT = """
You are an auditor. Count occurrences of the behavior DED (Dead-end recognition) in a
    competitive-programming reasoning trace.
DEFINITION
DED = Explicitly concluding the current approach is incorrect/insufficient or cannot
    meet constraints.
Include: naming a failure mode ("greedy not optimal", "breaks for duplicates", "TLE
    for n=2e5").

COUNT
- Count 1 per DED-labeled step.
OUTPUT (strict JSON ONLY):
{
  "DED": <integer count>,
  "events": [
    {"snippet": "<short quote>", "reason": "<why it matches DED>"}
  ]
}
<TRACE>
{TRACE}
</TRACE>

Analyze the trace and count the occurrences of DED.
"""
```

Prompt 5: DED_PROMPT (Dead-end Recognition)

```
BKT_PROMPT = """
You are an auditor. Count occurrences of the behavior BKT (Backtracking) in a
```

```
    competitive-programming reasoning trace.
DEFINITION
BKT = Revising or replacing the plan after recognizing a failure/limitation.
Include: "scrap/switch/replace", "instead we will...", "new plan: ...".
COUNT
- Count 1 per BKT-labeled step.

OUTPUT (strict JSON ONLY):
{
  "BKT": <integer count>,
  "events": [
    {"snippet": "<short quote>", "reason": "<why it matches BKT>"}
  ]
}
<TRACE>
{TRACE}
</TRACE>

Analyze the trace and count the occurrences of BKT.
{TRACE}
"""
```

Prompt 6: BKT_PROMPT (Backtracking)

```
AP_PROMPT = """
You are an auditor. Count occurrences of the behavior AP (Algorithm / Proof analysis)
    in a competitive-programming reasoning trace.

DEFINITION
AP = Justifying WHY the chosen algorithm/structure is correct/appropriate (proof
    sketches, invariants used as correctness arguments, reductions implying correctness
    ).
Include: exchange/optimality arguments, loop-invariant proofs, reductions with
    correctness justification, structural reasoning that ensures the property.
COUNT
- Count 1 per AP-labeled step.

OUTPUT (strict JSON ONLY):
{
  "AP": <integer count>,
  "events": [
    {"snippet": "<short quote>", "reason": "<why it matches AP>"}
  ]
}
<TRACE>
{TRACE}
</TRACE>

Analyze the trace and count the occurrences of AP.
{TRACE}
"""
```

Prompt 7: AP_PROMPT (Algorithm/Proof Analysis)

```
PSD_PROMPT = """
You are an auditor. Count occurrences of the behavior PSD (Pseudo implementation) in a
    competitive-programming reasoning trace.
```

```
DEFINITION
PSD = Presenting the algorithm as structured steps or pseudocode with control flow,
    without full code.
Include: numbered/indented outlines; loops/ifs; while/for; state updates in an
    algorithmic outline.

COUNT
- Count 1 per PSD-labeled step.

OUTPUT (strict JSON ONLY):
{
  "PSD": <integer count>,
  "events": [
    {"snippet": "<short quote>", "reason": "<why it matches PSD>"}
  ]
}

<TRACE>
{TRACE}
</TRACE>

Analyze the trace and count the occurrences of PSD.
{TRACE}
"""
```

Prompt 8: PSD_PROMPT (Pseudo Implementation)

### E.11. Reasoning Behavior Analysis Quality Check

E.11.1. MANUAL ERROR ANALYSIS

We select 10 reasoning traces for GPT-OSS-120B analysis and develop a web visualizer to clearly display the detected behaviors. Based on definitions of each behavior category, we manually inspect all 313 detected behaviors from these 10 reasoning traces and assign correctness labels accordingly. Additionally, we manually identify and annotate any behaviors missed by GPT-OSS-120B.

Through this careful manual validation, we find that 269 out of the 313 behaviors (86%) are correctly classified. The 44 misclassified behaviors primarily stem from misclassification errors—for instance, GPT-OSS-120B sometimes exhibits oversensitivity to code-like language, misclassifying simple "if-else" analysis statements as pseudo-implementation due to the misleading presence of phrases such as "implementation steps." Another common source of errors includes information loss due to chunking and occasional hallucinations of non-existent behaviors. Additionally, we identify 32 behaviors (approximately 10% relative to detected behaviors) that GPT-OSS-120B fails to detect. These missing behaviors are evenly distributed across all five behavior categories without distinct error patterns, consisting of both straightforward and challenging examples. We also show a reasoning trace example in Figure 14. Overall, the manual analysis confirms that GPT-OSS-120B effectively identifies a significant majority of behaviors with robust accuracy, demonstrating substantial reliability in capturing model reasoning behaviors.

E.11.2. INTER-MODEL AGREEMENT

To further strengthen the validation, we randomly sample 50 additional reasoning traces and pass them into Claude-Sonnet-4.5. Given the complexity and varying granularity of model annotations, instead of directly comparing text spans, we compute Krippendorff's alpha between the two models based on normalized behavior counts. Specifically, we calculate the percentage share of each behavior detected by models and compare their ranking across 50 traces. We observe the following inter-model reliability scores: analysis ($\alpha$=0.8018), planning ($\alpha$=0.7138), exploration ($\alpha$=0.8216), implementation ($\alpha$=0.6895), and verification ($\alpha$=0.6325). The two models show strong and consistent correlation for analysis and exploration behaviors, moderately strong agreement for planning and implementation behaviors, and moderate but slightly lower correlation for verification behaviors. These results further support the reliability and consistency of behavior classifications

performed by GPT-OSS-120B.

---

**GPT-OSS-120B-Medium's trace on USACO-2025-OpenContest-Sequence_Construction**

[1] **[Planning]** We need to construct sequence length up to 100 with sum M and xor of popcounts equals K.

[2] ... (7 lines omitted)

[3] **[Planning]** Goal: Find multiset of popcounts ($c_i$) ($0 <= c_i <= 30$) such that xor of $c_i$ = K and sum of some numbers with those popcounts equals M.

[4] ... (1 line omitted)

[5] **[Analysis]** However we can realize any popcount c can be realized by a number with value at most ($2^c - 1$) (e.g., lower bits).

[6] ... (86 lines omitted)

[7] **[Planning]** Thus problem reduces to: For given desired popcount c and target value S ($0<=S<=M$), does there exist a non-negative integer x with popcount(x)=c and x=S?

[8] ... (46 lines omitted)

[9] **[Analysis]** Thus we can start with binary decomposition of M into bits: we have list L of powers-of-two (size = popcount(M)).

[10] ... (19 lines omitted)

[11] **[Analysis]** If we split a number with popcount c into c ones: before popcount = c, after xor = (1 xor 1 ...

[12] ... (29 lines omitted)

[13] **[Implementation]** Implementation plan: - Decompose M into list of powers-of-two bits (each value = $2^i$). **[INCORRECT]**

[14] ... (28 lines omitted)

[15] **[Verification]** Let's test: starting X parity = (number of bits) %2 = popcount(M) %2.
    M=10 (1010) popcount=2, X=0.
    Target K=5 (101) delta=5 xor 0 =5 (odd).
    Since delta odd cannot be achieved by merging distinct bits (even changes).
    So maybe impossible, matching sample.

[16] ... (9 lines omitted)

[17] **[Verification]** Check other samples: M=2 (10) popcount=1 odd, K=1 odd -> same parity -> possible.

[18] **[Verification]** M=33 (100001) popcount=2 even, K=5 odd => different parity, but sample says possible!

[19] ... (33 lines omitted)

[20] **[Analysis]** If not found with 2 numbers, try 3 numbers: enumerate a,b from set S (size few thousand) and compute c = M-a-b, check if pc(c) = K xor pc(a) xor pc(b).
    Ensure c >=0.

This is $O(|S|^2)$ ~ few million per test, okay for 5k?

[21] ... (22 lines omitted)

[22] **[Implementation]** Idea: Use DP over bits to construct a with given popcount p such that popcount(M-a)=q. **[INCORRECT]**

[23] **[Analysis]** We can DP over bits with carry (borrow) state.
Define processing bits from LSB to MSB with borrow flag (0/1).
For each bit i, we decide $a_i$ (0/1).
Then $b_i = (M_i - a_i - \text{borrow}) \bmod 2$, and new borrow = $(a_i + \text{borrow} > M_i)$ ? 1 : 0.
We also count popcounts of a and b.
DP can compute if reachable.
We need to know if there exists a with pc(a)=p and pc(b)=q.
Since p,q <=31, DP state size: position up to 30, borrow 0/1, countA up to 31, countB up to 31.
That's 31*31*2*31=~60000 manageable.

[24] ... (11 lines omitted)

[25] **[Analysis]** Implement DP bool dp[pos][borrow][pa][pb] reachable.
Initialize dp[0][0][0][0] = true.
Iterate bits i from 0..30 (since $M < 2^{30}$).
At each step, known $M_i$ (0/1).
For $a_i$ in {0,1}.
Then compute diff = $M_i - a_i - \text{borrow}$.
$b_i$ = (diff & 1) (i.e., (diff+2)%2).
newBorrow = diff<0 ? 1 : 0.
Update counts.
At end pos=30, require borrow=0.
Then we have set of reachable (pa,pb).

[26] ... (2 lines omitted)

[27] **[Analysis]** Then we have set of reachable (pa,pb).

[28] **[Analysis]** If there exists pa such that pa xor pb == K, then we have solution with two numbers.

[29] ... (1 line omitted)

[30] **[Analysis]** We can also reconstruct by storing predecessor.

[31] **[Exploration]** If no such pair, we can try three numbers?

[32] **[Exploration]** If no such pair, we can try three numbers?

[33] ... (6 lines omitted)

[34] **[Exploration]** But we can extend DP to three numbers?

[35] ... (8 lines omitted)

[36] **[Exploration]** Simplify: Use DP to find any multiset of up to 5 numbers (since N<=100) with small popcounts by using DP over count of numbers?

Too heavy.
Given time, maybe we can produce algorithm: Use DP to find two numbers (a,b).

[37] ... (1 line omitted)

[38] **[Analysis]** Given time, maybe we can produce algorithm: Use DP to find two numbers (a,b).

[39] ... (12 lines omitted)

[40] **[Exploration]** if not found, output -1. **[INCORRECT]**

[41] ... (29 lines omitted)

[42] **[Analysis]** Complexities: dp size 31*2*32*32   63k bool, fine.

[43] **[Implementation]** Implement predecessor arrays as struct of 5 bits maybe small.
We'll store prev[bit][borrow][pa][pb] with struct containing prevBorrow, prevPa, prevPb, a_bit.
Also b_bit can be derived.
During transition we need b_bit which depends on M_i, a_bit, borrow.
We'll store a_bit only;
backtrack to compute.
Implement loops.
Edge: maximum popcount up to 30, but counts may go up to 31 (if all bits set).
Use 32 size.
Implementation details: int MAXB=31;
// bits 0..30 for (int i=0;i<MAXB;i++) { // i from 0 LSB int mi = (M»i)&1;
for borrow...
for pa...
for pb...
if dp[i][borrow][pa][pb] true for a_bit in {0,1} int diff = mi - a_bit - borrow;
... (29 lines omitted)

*Figure 14.* Behavior-labeled reasoning trace for the *Sequence Construction* problem (grouped behavior types shown in-line). We map low-level labels into five categories: **Planning** (PR/SUB), **Analysis** (AP/CMP), **Exploration** (BKT/DED), **Implementation** (PSD), and **Verification** (V–T). Misclassified events are marked with **[INCORRECT]**; across our evaluation on 10 reasoning traces, the 44 misclassifications primarily arise from (i) oversensitivity to code-like phrasing (e.g., "implementation steps" triggering pseudo-implementation), (ii) information loss due to chunking, and (iii) occasional hallucinated (non-existent) behaviors.

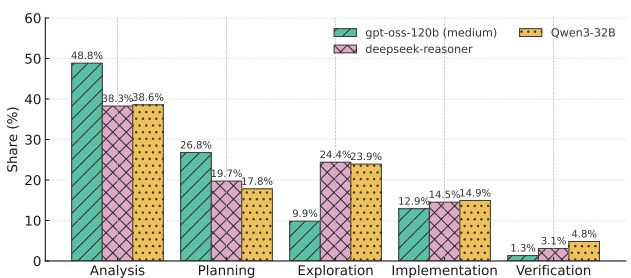 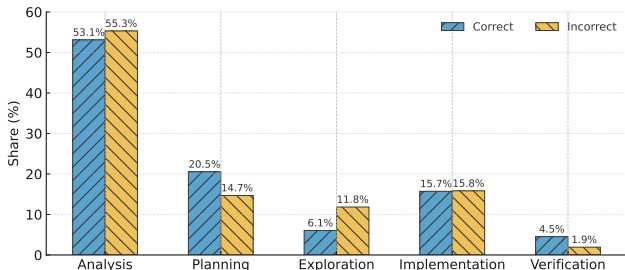

*(a)* Model comparison. Stronger reasoning models reduce unnecessary exploration, dedicating more resources to planning, structured analysis, and solution development.

*(b)* GPT-OSS-120B-high across correct/incorrect. Correct solutions depend heavily on initial structured planning and verification, reducing the need for exploration and continuous re-analysis.

*Figure 15.* **Reasoning Trace Analyses.** We categorize eight reasoning behaviors and divide them into five groups: **Analysis** (Algorithm/Proof analysis and Complexity Analysis), **Planning** (Problem Restatement and Subgoal Setting), **Exploration** (Backtracking and Dead-end recognition), **Implementation** (Pseudo implementation), **Verification** (Test Case Verification).

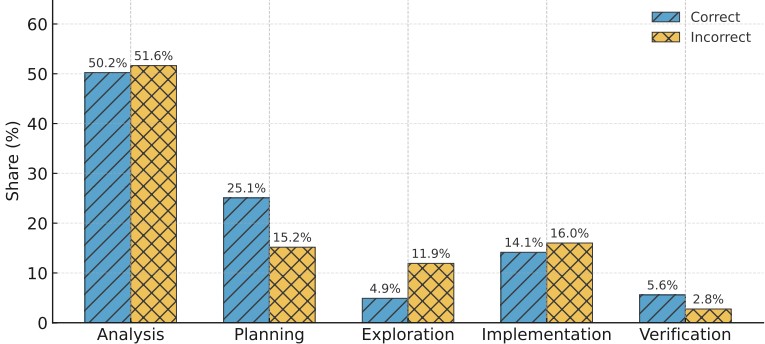

*Figure 16.* The reasoning behaviors of GPT-OSS-120B-High on easy problems across correct and incorrect solutions. Planning and verification behaviors are still important for models to produce correct solutions.

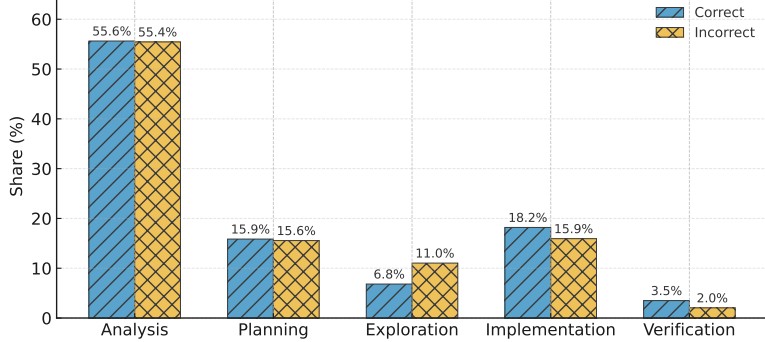

*Figure 17.* The reasoning behaviors of GPT-OSS-120B-High on medium problems across correct and incorrect solutions. Similar to easy problems, correct solutions show less exploration and more verification behaviors.

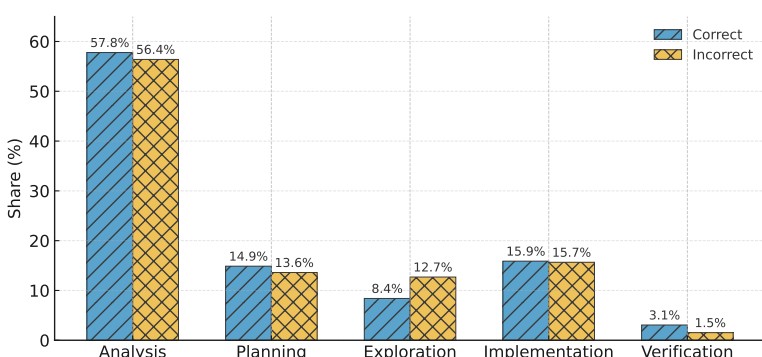

*Figure 18.* The reasoning behaviors of GPT-OSS-120B-High on hard problems across correct and incorrect solutions. Analysis, planning, and verification behaviors remain important for models to produce correct solutions.

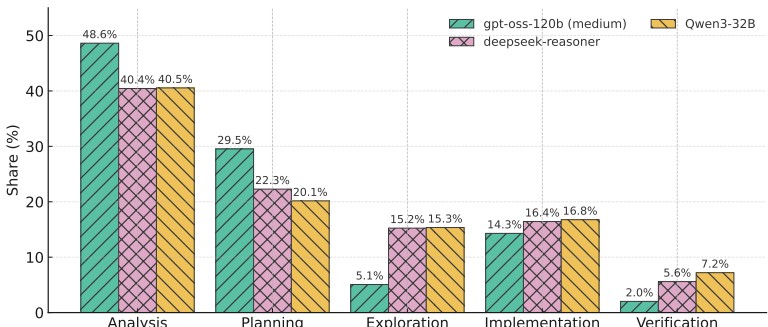

*Figure 19.* The reasoning behaviors of models producing correct solutions. Stronger reasoning models reduce unnecessary exploration, dedicating more resources to planning, structured analysis, and solution development.

