# OpenReview forum: "LiveOIBench: Can Large Language Models Outperform Human Contestants in Informatics Olympiads?"
_ICML.cc/2026/Conference — ICML 2026 regular_

### Official Review · Reviewer_DYTd · 2026-02-22

**Soundness:** 3
**Presentation:** 3
**Significance:** 3
**Originality:** 2
**Overall Recommendation:** 4
**Confidence:** 4

**Summary:**

This paper introduces LiveOIBench, a new benchmark for evaluating Large Language Models (LLMs) on competitive programming tasks from informatics Olympiads. Its core contributions are: (1) using expert-crafted, official test cases from 14 competitions to address the false positive issue in existing benchmarks; (2) introducing multi-faceted evaluation metrics including human percentile ranks and Codeforces ELO ratings; and (3) designing an automated pipeline for live updates to mitigate data contamination. The authors evaluate 34 LLMs and present a series of observational findings about model performance.

**Compliance With Llm Reviewing Policy:**

Affirmed.

**Final Justification:**

The paper presents a technically solid and practically valuable benchmark with high-quality test cases and a sustainable pipeline, but its scientific originality remains incremental and the interpretability of the core "Human Percentile" metric is insufficiently rigorous; the rebuttal partially addresses these concerns, warranting a weak accept contingent on explicit acknowledgment of the metric's limitations in the final version.

**Key Questions For Authors:**

- What is the methodological justification for aggregating human rankings from competitions of disparate levels (e.g., IOI vs. COCI) to compute a single global "Human Percentile"? How can this metric consistently reflect a model's capability relative to truly top-tier human competitors?

- Beyond code similarity analysis, are there plans to detect whether models have been exposed to the solution ideas or algorithmic discussions of these problems in their training data? Could this affect the attribution of performance to pure "reasoning capability"?

- Given that "live updates" may compromise the comparability of results across different time periods, how do you plan to manage versioning or provide static snapshots to ensure research reproducibility?

**Limitations:**

- Metric Rigor: The construction of the core "Human Percentile" metric weakens its interpretability.

- Boundary of Scientific Contribution: The work is fundamentally an integration and optimization of existing benchmarks, with limited scientific originality.

- Depth of Contamination Analysis: The study does not rule out the possibility of models succeeding by memorizing "problem-solving paradigms" rather than through pure reasoning.

- Language Limitation: The focus on C++ limits its applicability for evaluating models primarily trained on other languages.

- Temporal Scope: Restricting problems to post-2023 results in a relatively small dataset size.

**Strengths And Weaknesses:**

**Strengths:**

- High Data Quality: The use of official, expert-curated test cases is the benchmark's strongest asset, directly tackling a critical flaw in existing datasets and providing the community with a more reliable evaluation tool.

- Multi-Dimensional Evaluation Framework: Moving beyond a single pass@k metric to incorporate human performance as a reference point (percentiles, medals, ELO) offers a richer and more nuanced perspective on model capabilities.

- Engineering and Sustainability: The automated data crawling and maintenance pipeline ensures the benchmark's longevity and provides a practical, engineering-oriented solution to the data contamination problem.

**Weaknesses:**

- Logical Flaw in the "Human Percentile" Metric: The methodology of aggregating human rankings from competitions of vastly different difficulty levels (e.g., IOI vs. COCI) to compute a global percentile is methodologically questionable. This renders the final "Human Percentile" metric physically ambiguous—what does it truly mean for a model to surpass the 60th percentile? Does it signify approaching an IOI medalist or simply outperforming an average COCI participant? This cross-contest aggregation lacks statistical or behavioral justification.

- Incremental Rather than Foundational Scientific Contribution: The benchmark essentially integrates, scales, and optimizes the ideas from several recent, concurrent works (OI-Bench, HLCE, USACO). While it is larger and more up-to-date, it does not introduce a novel evaluation paradigm or reveal scientific insights that go beyond the current consensus. Its main findings (proprietary > open-weight, thinking > non-thinking) validate existing knowledge rather than providing a breakthrough.

---

> ### Author Rebuttal · Authors · 2026-03-31
>
> We thank the reviewer for the valuable feedback and constructive comments. Below we address the key concerns.
>
> ---
>
> ### W1 + Q1 + L1: Human Percentile Aggregation
>
> The purpose of aggregating human rankings is to provide a single, unified metric that distinguishes model performance across competitions. The reported “human percentile” reflects the average performance of a model across competitions.
>
> To address concerns about interpretability across varying difficulty levels (e.g., IOI vs. COCI), we provide more fine-grained analyses in Table 12 (Appendix), where we break down model performance across contests of different difficulty levels. In addition, our leaderboard website allows users to filter by specific competitions and directly examine model performance on selected subsets.
>
> ---
>
> ### W2 + L2: Scientific Contribution
>
> Beyond constructing a high-quality benchmark with complete test suites, subtask rubrics, and human contestant results, we conduct extensive analyses on:
>
> - Model performance across diverse algorithmic categories
> - Reasoning trace behaviors
> - Effects of data contamination
> - Solution submission outcomes
>
> We believe these analyses provide meaningful scientific insights that help the community better understand model behavior on competitive coding tasks and guide the development of more effective reasoning models.
>
> ---
>
> ### Q2 + L3: Contamination via Solution Paradigms
>
> We believe that this can be a valuable supplement to our current data contamination analysis.
>
> For a subset of contests, we collect official editorials containing solution explanations. Similar to Figure 3, we can measure model familiarity with these editorials and analyze its correlation with performance. However, given the low similarity between model-generated solutions and official solutions, we find it unlikely that models are memorizing solution ideas.
>
> If models had been exposed to official editorials, we would expect their implementations to more closely resemble the official solutions, which is not observed in our analysis.
>
> ---
>
> ### Q3: Benchmark Versioning and Reproducibility
>
> We plan to update LiveOIBench every 3–6 months to incorporate newly released models and contest problems. Each version will include both new contests and those from previous releases.
>
> To support reproducibility, we will maintain versioned releases of the benchmark, allowing researchers to evaluate models on fixed snapshots.
>
> ---
>
> ### L4: Language Choice (C++)
>
> All evaluations are conducted in C++, as it is the official language for most competitive programming contests and is widely preferred for its efficiency.
>
> In Appendix F.3, we compare GPT-OSS-120B’s performance across C++, Python, and Java on a subset of 132 USACO problems. We find that C++’s advantage primarily stems from computational efficiency rather than differences in reasoning capability.
>
> ---
>
> ### L5: Temporal Scope
>
> We restrict our dataset to post-2023 contests to minimize the risk of training data contamination. Older contests are more likely to have been included in model training corpora.
>
> We will continue expanding the benchmark over time through regular updates.
>
> ---

---

> > ### Author Rebuttal · Reviewer_DYTd · 2026-04-01
> >
> > The rebuttal improves our understanding of the benchmark's design choices, but the core concern about the rigorousness of the "Human Percentile" metric as a primary, globally reported figure remains. We maintain our overall recommendation at 4: Weak accept, with the suggestion that the authors add a more explicit discussion of this metric's limitations in the final version of the paper.

---

> > > ### Author Response · Authors · 2026-04-07
> > >
> > > Thank you for your reply. In the camera-ready version, we will add a discussion of “Human Percentile” metric to clarify that it is aggregated across contests with varying difficulty levels, and direct readers to the leaderboard website for more interpretable, per-contest results.

---

### Official Review · Reviewer_s1v9 · 2026-03-05

**Soundness:** 3
**Presentation:** 3
**Significance:** 3
**Originality:** 3
**Overall Recommendation:** 4
**Confidence:** 3

**Summary:**

This paper introduces a new benchmark, LiveOIBench, in response to issues such as low quality, insufficient difficulty, and lack of timely updates in existing code benchmark suites. The benchmark collects problems from Olympiad in Informatics competitions worldwide, resulting in 403 problems through web crawling, manual validation, PDF parsing, and metadata enhancement. After testing various open-source and closed-source models, the authors conduct analyses on model performance patterns on this benchmark, contamination control, and other aspects.

**Compliance With Llm Reviewing Policy:**

Affirmed.

**Final Justification:**

Overall, the quality of this work is good and may elicit some response from the community. Therefore I maintain my weak recommendation.

**Key Questions For Authors:**

1. To the best of my limited knowledge, Olympiad-level informatics competition benchmarks already exist, such as OIBench. I do not see significantly more effort in data collection in this paper compared to existing work. Moreover, since many OI complete test suites are not publicly available, how do the authors ensure that their test suites are official and public?
2. Test suites are central to the evaluation of code benchmarks. Given the high costs that hinder large-scale automation, have the authors made additional efforts or gained insights in improving and collecting test suites?

**Limitations:**

No need for discussion

**Strengths And Weaknesses:**

This work makes a substantial contribution. The authors have collected a significant amount of Olympiad in Informatics competition data, ensuring its quality through cleaning and validation processes. Additionally, the authors have conducted extensive in-depth analyses and testing on the benchmark, representing a meaningful contribution.

Weaknesses:

The authors mention that both the problems and test suites were obtained through web crawling. Does this imply that such data is publicly available, the collection cost is not particularly high, and it may have already been used in training existing closed-source models (though perhaps undetectable through current contamination monitoring methods)?

The potential impact may be limited. Given that research focus in code LLMs has gradually shifted from competition benchmarks toward solving practical software engineering problems, I have some concerns about whether this benchmark will generate broad interest in the code community.

---

> ### Author Rebuttal · Authors · 2026-03-31
>
> We thank the reviewer for the constructive feedback. Below we address the key concerns.
>
> ---
>
> ### W1: Contamination Analysis
>
> In Section 5.3, we conduct a comprehensive analysis of potential data contamination in our dataset.
>
> First, we evaluate whether model performance correlates with problem release dates, particularly around each model’s training cutoff. As shown in Figure 4, we observe no meaningful relationship between publication time and performance.
>
> Second, we examine whether model familiarity with task statements and official solutions impacts performance. Using GPT-OSS-120B, Qwen3-14B, and Seed-OSS-36B, we find near-zero correlation between familiarity and performance across all models. Please see our response to Reviewer RnyR for new Qwen3-14B and Seed-OSS-36B results
>
> Third, we use the source code plagiarism detection tool Dolos to measure similarity between model-generated solutions (from GPT-5 and Grok-4-Fast) and official solutions. GPT-5 achieves a median similarity of 0.11, while Grok-4-Fast achieves 0.12. As shown in Figure 9, the majority of solutions have similarity scores below 0.3, indicating that models are not simply memorizing templates.
>
> Additionally, we conduct an adversarial rephrasing experiment. We use GPT-5.4 to rewrite 71 problems from 2023 while preserving their semantics, and evaluate GPT-OSS-120B on both original and rephrased versions:
>
> | Model | Relative Score (%) | Human Percentile (%) | Pass Rate (%) |
> |------|--------------------|----------------------|---------------|
> | gpt-oss-120b-medium | 33.40 | 42.99 | 30.99 |
> | gpt-oss-120b-medium-rephrased | 33.36 | 42.57 | 30.99 |
>
> Performance remains unchanged, suggesting a very low probability of contamination through task statements.
>
> ---
>
> ### W2: Benchmark Contribution and Relevance
>
> While software engineering benchmarks have recently gained attention, we believe our benchmark provides complementary and valuable contributions:
>
> - High-quality Olympiad-level problems with complete test cases and subtask rubrics
> - Direct comparison to human contestants
> - Continuous updates with contamination controls
> - Integrated offline evaluation system
>
> ---
>
> ### Q1: Comparison with Existing Benchmarks
>
> We are the only competitive coding benchmark that collects subtask rubrics, full official test suites, and human contestant results.
>
> Existing benchmarks often suffer from incomplete test suites and rely heavily on platforms such as Codeforces for both data collection and evaluation, which limits accessibility and coverage. OIBench primarily depends on private, non-English contests and lacks continuous updates. Moreover, many prior benchmarks rely solely on pass rate as the evaluation metric, which provides a limited view of model performance.
>
> In contrast, our benchmark covers multiple Olympiads, supports continuous updates through direct collection of official materials, and incorporates richer evaluation metrics such as subtask scoring and human percentile comparisons.
>
> ---
>
> ### Q2: Data Source and Test Suite Reliability
>
> All well-known OI competitions release official materials, including complete test suites, on their websites. We collect data exclusively from official sources or from reputable online judging platforms when official websites are no longer maintained.
>
> To ensure scalability and consistency, we implement dedicated crawlers for each competition and platform. Our crawler code is included in the anonymous GitHub repository to support reproducibility.
>
> ---

---

> > ### Author Rebuttal · Reviewer_s1v9 · 2026-04-04
> >
> > I am grateful for the author's response. My primary concern is that, since the author was able to easily acquire the data using web crawlers, this data presents only two possibilities: either its quality is mediocre, or it has already been exposed to language models. In other words, I do not perceive any significant barriers to collecting this benchmark, nor do I see any unique characteristics; consequently, I am inclined to maintain my "Weak Accept" recommendation.

---

> > > ### Author Response · Authors · 2026-04-07
> > >
> > > Thank you for your reply. We respectfully disagree that public accessibility implies low quality or limited novelty. Our benchmark is curated from official materials of various OI contests, which are inherently high-quality and expert-designed. Many widely used benchmarks (e.g., AIME and LiveCodeBench) are also built from publicly available materials, yet are widely adopted due to their difficulty and expert curation. Our benchmark follows the same principle.
> > >
> > > In addition, we collect complete official test suites, subtask rubrics, and human contestant results, which are difficult to obtain due to fragmented formats across contest websites, where hidden tests and scoring logic are often not publicly structured (e.g., Codeforces does not disclose hidden tests). Prior work has considered using Codeforces problems, but typically stops at collecting problem statements and public test cases due to these challenges. In contrast, these full evaluation components are precisely what make our benchmark high-quality and enable rigorous, fine-grained assessment.
> > >
> > > Finally, our contamination analysis (Section 5.3) shows consistently negligible evidence of memorization, indicating that models are not simply recalling these problems despite their public availability.

---

### Official Review · Reviewer_RnyR · 2026-03-11

**Soundness:** 3
**Presentation:** 4
**Significance:** 3
**Originality:** 3
**Overall Recommendation:** 4
**Confidence:** 3

**Summary:**

Given the saturation of the current crop of competitive coding problem benchmarks, the authors propose LiveOIBench that covers Informatics Olympiads from 2023 to 2025. They curate problems from online sources spanning over 72 contests. The 403 problems average 60 test cases per and enable a direct comparison to elite human contestants. Finally they propose continuous to prevent contamination. Their findings show that GPT OSS 120B is the best open weight model at 60th percentile performance. GPT-5 produced the best proprietary model performance at 81.76th percentile.

**Compliance With Llm Reviewing Policy:**

Affirmed.

**Final Justification:**

The authors did provide more information regarding the contamination. However, I am now more unsure about the validity of their anti-contamination methods than I was prior. So I will keep the same score but lower my confidence.

**Key Questions For Authors:**

* For the open source models, how does the compute used impact the ranking of the models? For closed-source, is GPT-5 more cost-efficient?
* How do Opus 4.5 or GPT 5.2 perform on the benchmark?
* Is there a marked improvement if you formulate the problems as agentic coding with access to tools?

**Limitations:**

yes

**Strengths And Weaknesses:**

**Strengths:**
* A novel and well motivated benchmark that is needed for coding tasks. It additionally shows clear room left for improvement in both closed and open models.
* The paper is very well written and quite easy to follow. There are a few very minor nits I have:
	* Figure 1 is noisy with the zoom in and all of the annotations. Maybe it would be better to only annotate the top models and not zoom in.
	* Table 3: Pink is a very odd choice and hard to grep the gradient scale for what is better or not. I would recommend Red -> White -> Green.
* They have exhaustively benchmarked the frontier of models and provide great analysis of how each performs.
* The analysis through categories and on reasoning tokens is exceptionally well done and useful for the community.

**Weakness:**
The only weakness of this paper is that the contamination analysis is not convincing enough to alleviate fears that the model's performance is due to having seen these problems during training. Specifically, using GPT-OSS-120B solely for analysis is insufficient to confidently support their claims of no contamination. At the very least, it needs to be done on multiple models. Figure 4 better supports the claim than Figure 3; it should be moved to the main body and cleaned up slightly to improve legibility. Figure 3 is challenging to parse and less convincing. The Dolo analysis is a good finding in support of their claim, but it would be more convincing to adversarial change some of the problems (i.e., swap examples) from 2023, then look at how well the models do.

Overall, if you strengthen the anti-contamination argument and evidence, then this paper becomes a clear accept.

---

> ### Author Rebuttal · Authors · 2026-03-31
>
> We thank the reviewer for the thoughtful and constructive feedback. Below we address the key concerns and questions.
>
> ---
>
> ### Figure Presentation
>
> > “Figure 4 better supports the claim than Figure 3”
>
> In the camera-ready version, we will move Figure 4 to the main paper and improve the presentation of both figures for better clarity and readability.
>
> ---
>
> ### Contamination Analysis Across Models
>
> > “At the very least, it needs to be done on multiple models”
>
> We extend our contamination analysis to two additional models:
> - Qwen3-14B: https://ibb.co/P3dNm0V
> - Seed-OSS-36B: https://ibb.co/fbhFxRY
>
> For both models, we observe consistent trends with GPT-OSS-120B: there is no correlation between model familiarity with task statements or solutions and performance. This strengthens our claim that there is minimal data contamination in our benchmark.
>
> ---
>
> ### Adversarial Rewriting of Problems
>
> > “It would be more convincing to adversarially change some of the problems (i.e., swap examples) from 2023”
>
> We conduct an adversarial rephrasing experiment by sampling 71 problems from 2023 and rewriting them using GPT-5.4, preserving the original semantics while altering surface form. Due to time constraints, we only evaluate GPT-OSS-120B on both the original and rephrased versions:
>
> | Model | Relative Score (%) | Human Percentile (%) | Pass Rate (%) |
> |------|--------------------|----------------------|---------------|
> | gpt-oss-120b-medium | 33.40 | 42.99 | 30.99 |
> | gpt-oss-120b-medium-rephrased | 33.36 | 42.57 | 30.99 |
>
> Performance remains effectively unchanged, suggesting a very low likelihood that models rely on memorized task statements.
>
> ---
>
> ### Q1: Impact of Compute on Model Performance
>
> In Figure 5 (Appendix), we show that model performance improves with increased compute budget under both parallel and sequential scaling regimes. Notably, GPT-OSS-20B with a high reasoning budget outperforms GPT-OSS-120B with a medium reasoning budget. Similarly, Table 2 shows a substantial performance gap between thinking and non-thinking modes for Qwen-3 models. These results indicate that reasoning budget plays a critical role, and smaller models can outperform larger ones when given sufficient compute. Among closed-source models, GPT-5 is the most cost-efficient, achieving strong performance with fewer tokens.
>
> ---
>
> ### Q2: Performance of Newer Models
>
> We evaluate GPT-5.4 on our benchmark using the default reasoning budget:
>
> | Model | Gold (%) | Medals (%) | Relative Score (%) | Human Percentile (%) | Pass Rate (%) | ELO |
> |------|----------|------------|--------------------|----------------------|---------------|-----|
> | GPT-5.4 | 72.22 | 91.67 | 68.30 | 84.69 | 76.25 | 2507 |
> | GPT-5 | 50.00 | 88.89 | 67.21 | 81.76 | 63.03 | 2414 |
> | Grok-4-Fast-Reasoning | 45.83 | 83.33 | 56.99 | 74.23 | 50.95 | 2221 |
> | Gemini-2.5-Pro | 31.94 | 77.78 | 51.33 | 71.80 | 44.46 | 2192 |
> | GPT-O3-Mini-High | 26.39 | 72.22 | 47.69 | 64.28 | 44.19 | 2088 |
> | Gemini-2.5-Flash | 15.28 | 62.50 | 41.29 | 56.81 | 36.06 | 1945 |
> | Claude-Sonnet-4.5 | 11.11 | 66.68 | 38.30 | 53.08 | 27.05 | 1848 |
> | GPT-4.1 | 4.17 | 40.28 | 24.78 | 35.99 | 18.32 | 1482 |
>
> ---
>
> ### Q3: Agentic Coding with Tools
>
> Yes. The OpenAI blog (OpenAI, *Introducing o3 and o4-mini*) shows that providing models with tool access (e.g., terminal interaction) can significantly improve performance. However, in this work, we focus on the model’s intrinsic problem-solving skills on competitive coding questions without external tools. This allows for a more controlled assessment of core reasoning capabilities.
>
> ---

---

> > ### Author Rebuttal · Reviewer_RnyR · 2026-04-02
> >
> > Thank you for the response. The presentation comments have been fixed as well as the frontier models and coding tools. On the contamination side, I am left even more unconvinced by the analysis than before, and I do not think it is something that can be solved during this review period. The adversarial rewriting should not be done with GPT 5.4. Instead, it should be done by selecting different adversarial examples for the prompts. The rewriting performed just adds in more confounds rather than clearing up the contamination concerns.

---

> > > ### Author Response · Authors · 2026-04-07
> > >
> > > Thank you for your reply. We believe there is a misunderstanding between our rephrasing experiment and the reviewer’s proposed adversarial experiment.
> > >
> > > For our rephrasing experiments, the goal is not to introduce adversarial changes, but rather to generate semantically equivalent task statements that models are unlikely to have seen during training. To change the surface form without altering the semantics, we use a strong model like GPT-5.4 for rewriting.
> > >
> > > Below is an example snippet from a USACO problem:
> > >
> > > **Original snippet:**
> > > Pareidolia is the phenomenon where your eyes tend to see familiar patterns in images where none really exist—for example, seeing a face in a cloud. As you might imagine, with John’s constant proximity to cows, he often sees cow-related patterns in everyday objects. For example, if he looks at the string "bqessiyexbesszieb", John’s eyes ignore some of the letters and all he sees is "bessiebessie".
> > >
> > > **Rephrased snippet:**
> > > In a deep-space signal archive, analysts search noisy transmissions for repeated occurrences of the beacon code "orion7". A transmission may contain many extra symbols, and analysts are allowed to ignore some of them. For example, from the string "xoriqon7morionk7z", they can ignore certain characters and perceive only "orion7orion7".
> > >
> > > If models perform similarly on rephrased statements, this suggests they are not relying on memorization but instead solving the problems through reasoning.
> > >
> > > Beyond GPT-OSS-120B, we additionally evaluate two other model families. Below, we report relative scores:
> > >
> > > | Variant     | GPT-OSS | Seed-OSS | Qwen3-30B-A3B |
> > > |-------------|--------:|-----:|--------------:|
> > > | original    | 33.40   | 32.44| 21.09         |
> > > | rephrased   | 33.36   | 33.50| 20.99         |
> > >
> > > All models maintain their performance on rephrased statements, indicating a low likelihood of data contamination.
> > >
> > > ---
> > >
> > > In addition, we conduct adversarial change experiments aligned with the reviewer’s suggestion. The goal is to perturb task statements such that performance should degrade if models have not memorized the original problems.
> > >
> > > On the same set of 71 problems, we consider two types of adversarial modifications:
> > > - Removing input/output examples
> > > - Randomly masking tokens in the task statements
> > >
> > > We report results for GPT-OSS-120B, Seed-OSS-Instruct, and Qwen3-30B-A3B:
> > >
> > > - Δ (%): mean relative-score change compared to original task statements
> > > - d_z: paired standardized effect size (negative indicates degradation)
> > >
> > > | Variant          | GPT-OSS Δ (%) | GPT-OSS d_z | Seed-OSS Δ (%) | Seed-OSS d_z | Qwen3-30B Δ (%) | Qwen3-30B d_z |
> > > |------------------|-------------------:|-----------------:|---------------:|-------------:|----------------:|--------------:|
> > > | Remove examples  | -5.24%             | -0.13            | -6.58%         | -0.27        | -7.32%          | -0.23         |
> > > | Mask 10%         | -4.63%             | -0.14            | -2.32%         | -0.12        | -5.25%          | -0.47         |
> > > | Mask 30%         | -8.59%             | -0.21            | -17.44%        | -0.45        | -30.98%         | -0.56         |
> > > | Mask 50%         | -38.58%            | -0.89            | -28.88%        | -0.52        | -38.33%         | -0.69         |
> > >
> > > We observe that removing examples leads to a moderate performance drop, suggesting models are not relying on memorized examples. However, modifying examples alone is insufficient as a strong adversarial change, since capable reasoning models can still infer solutions without them, or even with incorrect ones.
> > >
> > > In contrast, masking tokens introduces stronger perturbations that degrade semantic clarity. If models had memorized the problems, they would be expected to recover performance. Instead, we observe consistent degradation as the masking ratio increases. At higher masking levels (30%, 50%), even strong reasoning models experience significant drops.
> > >
> > > ---
> > >
> > > We also want to re-emphasize our existing contamination analysis in Section 5.3:
> > >
> > > - **Temporal analysis:** We evaluate whether performance correlates with problem release dates relative to model training cutoffs. As shown in Figure 4, we observe no meaningful relationship.
> > > - **Familiarity analysis:** Using GPT-OSS-120B, Qwen3-14B, and Seed-OSS-36B, we find near-zero correlation between familiarity (task statements and solutions) and performance.
> > > - **Solution similarity:** Using the Dolos plagiarism detection tool, we compare model-generated solutions (GPT-5, Grok-4-Fast) with official solutions. Median similarity scores are low (0.11 and 0.12), with most below 0.3, indicating models are not memorizing templates.
> > > - **Template analysis (Appendix E.4, E.7):** We find only weak or negligible correlations (r ≈ −0.08 to 0.19) between solution similarity and performance, suggesting minimal template-based contamination.
> > >
> > > Together with the two additional experiments, we believe we provide a comprehensive analysis of potential data contamination in our benchmark.

---

### Official Review · Reviewer_5dQL · 2026-03-13

**Soundness:** 3
**Presentation:** 4
**Significance:** 3
**Originality:** 3
**Overall Recommendation:** 4
**Confidence:** 4

**Summary:**

The paper introduces LiveOIBench, a benchmark based on recent informatics olympiad contests, and argues that prior coding benchmarks often suffer from incomplete test suites, coarse difficulty granularity, reliance on online APIs, and overly coarse metrics. They select 403 expert-curated problems, averaging 60 official test cases each, drawn from 72 contests across 14 Informatics Olympiads held between 2023 and 2025.
The authors package
1) official/private-style test sets, subtask rubrics,
2) Continual update to reduce contamination risk
3) comparison to expert human results
4) A fully offline reproducible evaluation system,

They then evaluate 34 models and add analyses of algorithm categories, inference-time scaling, error types, reasoning traces, and contamination.

They find that GPT-5 achieves an average human percentile of ~82%, while exhibiting remarkable token efficiency with Grok-4-fast and Gemini-2.5-pro following it. Among open-source models, GPT OSS, Seed OSS, and Qwen 3 demonstrate progress in narrowing the gap with proprietary models.

**Compliance With Llm Reviewing Policy:**

Affirmed.

**Final Justification:**

My overall recommendation for this paper is a weak accept. A central and strong claim of the abstract and the paper at large is behind a weak evaluation setup. 1/3 of the dataset lacks supporting evidence for its central claim.

Reasons to accept the paper include extensive experiments across both open & closed models and a structured dataset.

**Key Questions For Authors:**

- The appendix says that when official websites are incomplete, the authors pull missing pieces from CSES, LibreOJ, and GitHub, including user-submitted accepted solutions. That is understandable, but it means the paper should distinguish “official-only” from “official-plus-curated-secondary-sources.” Would you agree?
- What exactly is the denominator for the human percentile? Is it averaged over 46 contests with human results, a subset of tasks, or imputed in some way for the full 72-contest table?

**Limitations:**

- The traces are chunked and annotated by GPT-OSS-120B itself, and the appendix (E.11.1) shows noticeable annotation error/misclassification in the manual check.
- Around 1/3rd of the benchmark lacks subtask information because of USACO.

**Strengths And Weaknesses:**

Strengths:
- The benchmark is of high use: Official contest sources, a local judge, and expert human comparisons make it much more valuable than a simple pass@1 benchmark.
- The empirical study is quite broad, with the evaluation of 34 models (both open and closed) on the benchmark.
- Gold/silver/bronze thresholds, human percentile, and Codeforces-style Elo are more interpretable than raw pass rate alone.
- The runtime-error breakdown and the claim that stronger reasoning models exhibit reduced exploration, allocating more resources toward solution development and analysis, are interesting and potentially actionable.
- The paper also has a lot of analysis studying inference time scaling, both parallel and sequential, performance variation across languages, and analyzing model performance with respect to different attributes.
- The data contamination study is useful, especially the fact that official solutions and model solutions differ in nature.

Weaknesses:
- Can LLMs outperform human contestants is a core question of this work, but its main evaluation allows 8 sampled solutions per problem and picks the highest-scoring one. That is a capability estimate, not a strict contest-style one-shot comparison.
- The human-comparison story is unclear to me. The appendix states that only 46 of 72 contests include human results, yet the main tables present “human percentile” results alongside the overall 72-contest evaluation.
- The subtasks' contribution is real, but not universal across the dataset. Table 7 says USACO contributes 132 tasks and explicitly notes that USACO doesn’t provide subtask information. There is no clarity on whether it's absent or whether subtasks have been added.
-  Some metadata is partially LLM-generated; it introduces annotation noise and should be emphasized more clearly when interpreting per-tag conclusions.

---

> ### Author Rebuttal · Authors · 2026-03-31
>
> Thank you for your positive feedback and comments. We aim to address your problems and concerns below:
>
> ### **W1**
> In standard OI contests, contestants are allowed to submit up to 50 solutions per problem. However, given resource constraints and the diminishing performance gains observed in Figure 5(a), we follow prior work and limit models to 8 submissions.
>
> ---
>
> ### **W2**
> Since USACO does not disclose contestant results, human percentiles are computed based on the 46 contests where such data is available. We will clarify this explicitly in the caption of the camera-ready version.
>
> ---
>
> ### **W3 + L2**
> USACO problems do not provide subtask information or human contestant results. Therefore, USACO is excluded from metrics that rely on these signals, including Human Percentile and Codeforces Rating.
>
> ---
>
> ### **W4**
> The majority of problems include algorithmic tags. Only a small portion of metadata is generated by LLMs, and we will clarify this more explicitly in the camera-ready version.
>
> ---
>
> ### **Q1**
> When official websites are incomplete, it is typically due to poor maintenance (e.g., broken download links). However, many online judging platforms archive these materials shortly after release. All materials used in our benchmark—including problem statements, test cases, and subtasks—are official. The only non-official components are user-submitted solutions, which are used solely to validate our evaluation judge.
>
> ---
>
> ### **Q2**
> Human percentile is computed using the 46 contests with available human results and is macro-averaged across 14 competitions.
>
> ---
>
> ### **L1**
> We acknowledge that annotation errors may arise from GPT-OSS-120B. In Appendix F.9, we conduct a manual error analysis on 10 reasoning traces and find that 269 out of 313 behaviors (86%) are correctly classified. Additionally, 32 behaviors (10% relative to the total) were missed.
>
> We further compare its annotations with those from Claude-Sonnet-4.5 on 50 reasoning traces and compute Krippendorff’s alpha, observing an average inter-model agreement of 0.7314 across five categories.
>
> Overall, these results indicate that GPT-OSS-120B reliably captures the majority of reasoning behaviors.

---

> > ### Author Rebuttal · Reviewer_5dQL · 2026-04-04
> >
> > I thank the authors for addressing all my questions.
> >
> > I believe a central claim in the abstract is "GPT-5 achieves an 81.76th percentile, and GPT-oss reaches 60th percentile", but this calculation is done over "46 contests with available human results and is macro-averaged across 14 competitions," and USACO is excluded. But USACO accounts for over 1/3 of the dataset, as clear from Table 7 & Table 8 in Appendix A. From what I understand, in the same table 2, certain columns use the full dataset while others do not? I think this should have been highlighted more clearly; the fact that one-third of the benchmark is not used for the claim in the abstract is an oversight.
> >
> > I am hence inclined to maintain my current score.

---

> > > ### Author Response · Authors · 2026-04-07
> > >
> > > Thank you for your reply. In the camera-ready version, we will clarify that human percentile is computed using contests with available human contestant results, both in the abstract and the main body.

---

### Decision · Program_Chairs · 2026-04-30

**Decision:**

Accept (regular)

**Comment:**

This paper introduces LiveOIBench, a new competitive programming benchmark built from official Informatics Olympiad contests, with full test suites, subtask-based scoring, offline evaluation, and direct comparison against human contestants.

This article’s central idea is that coding evaluation for modern LLMs now requires a more rigorous and realistic benchmark than prior platform-based or incomplete-test-suite settings, and that Olympiad-style tasks with official test cases and human score distributions provide a stronger foundation for measuring reasoning-heavy code generation. Overall, the authors examine a major issue in current coding evaluation: many existing benchmarks either underestimate difficulty, suffer from false positives due to incomplete hidden tests, or rely on online infrastructures that limit accessibility and reproducibility. The proposed benchmark directly targets these weaknesses through official public test data, fine-grained rubrics, and local judging.

All four reviewers are positive overall and recommend weak acceptance, and I agree with that consensus.